# Sustaining Human Resources through Talent Management Strategies and Employee Engagement in the Middle East Hotel Industry

Fida Hassanein [1,*] and Hale Özgit [2]

1   Department of Business Administration, Institute of Graduate Research and Studies,
    Cyprus International University, Nicosia 99258, North Cyprus, Turkey
2   School of Tourism and Hotel Management, Cyprus International University,
    Nicosia 99258, North Cyprus, Turkey
*   Correspondence: fidaraghebhassanein1980@gmail.com

**Abstract:** This study aims to develop a deeper understanding on talent management strategies to encourage employee engagement in the tourism industry of the Middle East during and after COVID-19 and further to reveal the impacts of employee engagement on customer satisfaction. This study was conducted qualitatively, where open-ended questions were posed to 37 managers through semi-structured interviews. Managers in the human resource domain of numerous hotels across the Middle East located in countries such as Lebanon, United Arab Emirates, Egypt, Jordan, Bahrain, Qatar, Saudi Arabia, Turkey and the Sultanate of Oman participated in the study. The majority of Hotels had five-star ratings, and others four-star. The study themes were qualitatively developed from the data using inductive content analysis deployed in QSR NVivo. The results showed that by implementing appropriate talent management strategies, engagement and, consequently, job satisfaction of hotel staff could be enhanced. The COVID-19 pandemic showed that realistic targets must need be set for the effective retention of talented employees. The results imply that a lack of resources and investments in talent management strategies (e.g., reward system) can lead to the loss of talented employees. The overarching impact of talented employees is increased customer satisfaction as service quality is improved and interactions between staff and clients are enhanced. The results are beneficial for scholars as well as leaders in the hotel industry of the Middle Eastern region.

**Keywords:** talent management; employee engagement; hotels and hospitality; Middle East; job satisfaction; qualitative analysis



## 1. Introduction

Talent management has evolved into a contemporary concept focusing on talented employees and the processes organizations implement to ensure that qualified people are attracted to and hired by the company. In the context of talent management, it is also essential to have strategies that facilitate the professional development of employees in order to acquire useful skills [1]. The current research focuses on talent management strategies in the context of the Middle Eastern hotel industry that, compared to other regions, is less studied. As talent management practices focus on organizational success [2], crises (i.e., COVID-19 pandemic) can hinder organizational processes and functions that, subsequently, reduce the effectiveness of these practices. Organizations have invested heavily in improving talent management methods as a primary function of the Human Resource Management (HRM) department. In this study, the focus is on the Middle Eastern hotel industry and how the pandemic affected HRM from the perspective of managers. Talented staff are key for companies based on the Pareto principle, where 20% of staff can aid the firm in achieving 80% of its goals [3] as the demand for skilled and talented individuals among businesses increases, a talent shortage in the employee pool can be

observed, highlighting the importance of talent management to the long-term viability of businesses [4,5]. This study focuses on the hotel industry within the tourism and hospitality industries, where success is measured by the quality of service, accessibility, and breadth of services offered to customers [6–8]. This necessitates service providers who are skilled and devoted in their roles as service providers. While managers in the hotel industry focus on attracting and recruiting talented employees, the Middle East has a dire need for a talent pool. Therefore, talent management extends beyond recruitment to include fostering employee engagement and satisfaction [6,7,9] in order to retain talent. In "normal" circumstances, talent management strategies are structured as described above. However, these strategies need to be better understood when confronted with a crisis; filling this gap is the focus of this study. It is important to note that hotels have struggled to retain their employees due to low wages, long hours, and a lack of long-term contracts. It is important to note that hotels have faced issues in retaining their staff due to low wages, long working hours, and lack of long-term contracts. The aforementioned issues, combined with the impacts of the pandemic, have hindered the engagement of employees and their satisfaction, which are emphasized in this research. This is linked to changes in work settings, lack of adequate resources and support, and increased demands of jobs that influence the perception and attitude of employees towards their organization. These aspects are highlighted in this study to note that talent management strategies were used by managers during the pandemic to ensure engagement and satisfaction for skilled staff. Therefore, it is essential that talent management is better understood during crises so that organizations can have a higher level of resilience, preparedness, and strategies that can help with retaining talent pool. As hotels were severely impacted during the pandemic, many employees were forced to leave their jobs, which poses threats to organizational performance and service quality (after businesses go back to their normal routines). This research endeavors to highlight talent management strategies that were deployed by hotel managers during the pandemic to yield in beneficial results for future decision-making for retaining talented staff while maintaining high quality services and performance. Providing support through appropriate leadership practices, especially during a crisis is also vital in this regard [4,5,10,11].

Employee engagement is each individual's effort and investment of their time into adapting new skills, gaining new knowledge and increasing creativity [9]. There is a direct relationship between employee engagement and talent management within a workplace which acquires and supports higher levels of skills and knowledge and aims for enhancing job satisfaction [9,10,12]. This research focuses on the importance of talent management strategies that emphasize on engagement and satisfaction of employees in the context of hotels as they are in constant interaction with customers and thus, can enhance customer experience, generate revenue, and provide value [13]. In the aftermath of COVID-19, talent management strategies and practices become more vital particularly, after the hospitality sector ceased its activities, which led to unemployment, impacting employees and their satisfaction and engagement [14]. As talent management strategies are directly linked to organizational support, employees' positive behaviors such as, engagement have drastically decreased during the pandemic. This has vivid the impacts on objective and subjective success of employees pertaining to promotion (professional development) and career outcomes (i.e., job satisfaction) [15–17]. As employees are aware of the qualities and intents that they possess, communication of high-quality information becomes critical within the context of talent management. Human capital science and theory are utilized, which emphasize the significance of meeting employee needs. These theories pertain to organizational strategies and approaches aimed at enhancing employees' work environments. As such, they apply to HRM practices that emphasize retaining talented employees by recognizing their needs. This research also incorporates the job demand–resources model, which describes the tangible and intangible resources available for talented personnel in the hotel industry, which is highly demanding and frequently lacks adequate resources. In this sense, this theory demonstrates that hotel employees, especially talented ones, require the necessary tools

and support to perform well. The theory of conservation of resources is also significant in this regard, as it pertains to the tendency of individuals to conserve their resources when confronted with demanding jobs and issues such as the pandemic that can deplete their mental and physical resources. In this regard, this theory aligns with the purpose of this study for a better understanding of strategies that can assist talented employees in the workplace and, consequently, assist the organization in retaining its talent pool. Consequently, the current study contributes to the application and comprehension of the aforementioned theories and existing literature, as well as to managerial decision-making in hotels regarding talent management strategies. This research of several aspects about talent management, including (1) the importance of employee engagement in the hotel industry; (2) talent management aspects (i.e., retention, evaluation, and overall hotel talent management strategies); (3) the role of managers in handling talent; and (4) the organizational setting in terms of talent management. These aspects are derived from the themes high the research's analysis section (see Table 1). Through understanding the aforementioned themes from the perspective and experiences of managers, this research aims to contribute to the literature on talent management, particularly in times of crisis. Therefore, employees' engagement, talent management practices, managerial role, and organizational setting to a better understanding of influential aspects of talent management in the hotel sector. The firsthand data obtained from several hotel managers can be beneficial for scholars and practitioners as it highlights tactics and strategies used during the pandemic to retain skilled staff.

**Table 1.** Themes of the research.

| | |
|---|---|
| **Theme 1** | Employee engagement |
| **Theme 2** | TMS in hospitality sector |
| **Theme 3** | Talent evaluation and talent retention |
| **Theme 4** | The role of managers in managing talented employee |
| **Theme 5** | Organizational culture and talent management strategy |

During the global pandemic, which severely affected all industries and the hotel industry, the Middle East continued to exhibit resilience in hotel performance. This has had negative effects on talent management and decreased hotel performance. On a global scale, the number of employees was reduced, operations slowed down or ceased entirely [18], and the sector continued its limited regional operations [19]. Consequently, the purpose of this study is to investigate and gain a deeper understanding of talent management strategies among Middle Eastern hotels and how they promote employee engagement during the COVID-19 pandemic. This study argues that in order to better manage the work outcomes of hotel employees during a crisis (i.e., the COVID-19 pandemic), talent management strategies designed to increase employee engagement and job satisfaction should be implemented. This can result in positive organizational outcomes such as service quality, increased performance, customer satisfaction, and increased revenue, which are essential for service sector organizations.

In the subsequent sections, the research methodology and relevant literature pertaining to talent management, as well as the significance of employee engagement, are discussed. This includes evaluating the attitudes and engagement of employees, as well as the problems that arise in this context. This is followed by a discussion of talent management strategies in the hospitality industry, including the skill requirements of employees and the challenges associated with retaining talent. In addition, the evaluation and retention of talent, as well as the benefits of talented employees, are discussed. The interview analysis and findings follow the literature immediately. Finally, the contribution of the current study is demonstrated by its implications for practitioners and future research.

## 2. Literature Review

*Talent Management Strategies and Employee Engagement*

Strategic talent management can be defined as the methodological identification of crucial jobs that directly contribute to the competitive advantage of an organization sustainably [20,21]. In the premise of human capital science and theory [22], talent management expands to the recognition and creation of processes that address the needs of employees [14,23], not only the needs and goals of the organization. This further establishes a talent pool with high potential that leads to a distinctive human resource structure through high commitment levels [24]. In this respect, talent management strategies distinguish decisions made in the HR department from its typical practices and planning by emphasizing skillfulness and diversity of individuals via identification, development, engagement, and retention [14,25]. Accordingly, the scope of talent management strategies comprehensively incorporates employment. It is also important to recognize the essential role of leaders (e.g., ethical) in establishing a non-toxic environment for employees that encourages positive behaviors (i.e., engagement) by focusing on their needs, and providing necessary support [10]. Through such strategies, the overall performance of organizations can be greatly enhanced [26]. During 'normal' times, talent management strategies follow the aforementioned framework, with managers and HR departments constantly seeking new talent while putting effort into developing the skills of existing staff [25]. However, there is a scarcity of qualitative studies [21] that gather in-depth data from managerial levels that can highlight challenges, and strategies that are undertaken (especially during the pandemic). Incidents, such as the pandemic, impact work processes that include talent management and require managers to implement tactics and strategies that fit the scale and scope of the current situation. As the current research is deployed qualitatively, the theoretical settings related to talent management strategies and their application in the context of tourism and hospitality are focused. Moreover, a gap is also noted regarding talent management and employee engagement during the pandemic. Due to the aforementioned issues that occurred in the hotel sector because of the pandemic, there is a gap in terms of understanding the issues of talent management in hotel sector [18,19,21]. This study combines the aforementioned gaps to contribute to the existing knowledge.

In the context of this research, it is important to note that a major driver for its conduct is the fact that the majority of studies have examined this subject in other regions (e.g., Southeast Asia, Western Europe, and Northern America), leaving Middle East relatively less examined [26,27]. As mentioned earlier, there needs to be more literature on talent management in hotels during the pandemic. Following the concept of talent management and its implications in the hotel sector of the region, this study argues that in the absence of adequately structured talent management procedures, the hospitality industry runs the risk of employing incompetent and unqualified employees. Hence, talent management strategies can play an essential role in maintaining competitive advantage through adjusting to structural, demographic, and environmental changes (i.e., in response to the pandemic). Subsequently, significant implications for employment security, working conditions, and other related aspects can be drawn from examining this subject. These strategies are effective and efficient when the business is active routinely [14]. However, crises such as the global pandemic which affects both physical and psychological domains of life, can force managers to take new measures for talent management as 'routine' procedures fail to persist. Notably, a limited number of studies have addressed talent management and its relevant factors during crises (e.g., [21,26,27]). This is important due to the decreased rate of travelling, borders closing, and limited exchange of tourism during the pandemic. This study focuses on the importance of employee engagement in this context as a crucial element in the tourism industry that is not been thoroughly examined during the COVID-19 outbreak. In this regard, this research highlights the vitality of understanding the challenges faced by managers in hotels in terms of talent management strategies that have been disrupted during the pandemic. The in-depth data that is derived from managers' perspectives can be beneficial for scholars and practitioners in tourism as it sheds light on

approaches and tactics that help hotels in retaining their talented staff. Hence, organizations can implement such strategies to increase their resilience and preparedness when facing future crises.

HR department, alongside adequate leadership in the hotel sector is essential for both internal and external aspects (e.g., employee engagement and customer experience) [10,11]. These lead to other positive work outcomes highly beneficial for the organization such as, innovative work behavior, job satisfaction, higher performance rate, and service quality [15,28,29]. Employees' engagement with their work is particularly important for the service industry due to constant interactions among staff and customers [13]. Importantly, as careers of individuals have been threatened during the pandemic, it becomes essential to encourage engagement to ensure that organizations can go back to their normal routines and enhance their success rate. In this sense, both physical and psychological domains are involved in this context [11]. Notably, high engagement levels for employees lead to exhibition of positive work behaviors and connectedness to the organizations' goals (e.g., productivity, and extra-role behaviors such as citizenship and proactive) [11,16,30]. Therefore, employee engagement is an essential aspect that organizations and their HR teams, and leadership approaches should emphasize on, and create a workplace, where engagement is fostered. Regarding talented employees, engagement is vital as it promotes positive behaviors in workplace and in interactions with customers and other members of the company. This becomes more vivid pertaining to the service and hotel sector. This is further linked to the job resources (that are not always sufficient in the hotel sector) and job demands (which are at high levels in hotels) [31–33], which pose direct influence on employees' work outcomes [30,34]. Hence, a theoretical model that encompasses these hardships is used in this research to address the demands and resources available for hotel staff. In this research, it is conceptualized that the pandemic limited the resources available for talented staff, and increased demands related to work (e.g., lower wages and lack of proper compensations or developmental career opportunities), which falls within the boundaries of JD-R model [11,33].

Within the hospitality sector, it is vital that organizations are equipped with employees who are engaged with their jobs due to the aforementioned high extent of interaction with customers [19,35]. This requires an active HR department emphasizing enhancing work settings for employees [11,30,31]. Building on the job demands-resources model (JD-R) [36,37], this research focuses on practices of HR particularly, in the talent management context regarding enhancing work engagement of employees in hotels across the Middle East. Positive outcomes can be achieved by recognizing tangible and intangible aspects of the job and focusing on providing necessary means (resources) for skilled employees [11,38–40]. JD-R fits in the current context as employees in hotel sector require numerous resources due to constant interactions and other hardships (e.g., long hours, low wages and short-term contracts). Thus, managers of hotels and their HR managers have been addressed in terms of their approaches and initiatives towards this subject. Within the premises of the JD-R model, the needs of employees can and should be focused on so that a balance is obtained and, thus, engagement is increased [40,41]. The JD-R model fits this context as it aligns with the disruptions in HR strategies caused by the pandemic (i.e., limitations of resources, and changes in job demands). The aforementioned issues caused by the pandemic decreased engagement as it jeopardized the physical and psychological resources of individuals while increasing the demands of their jobs (i.e., lower income). Hence, this research can contribute to the current understanding of this model in the context of hotel talent management strategies during the pandemic.

In addition to what was mentioned above, the Conservation of Resources (COR) theory is also embedded in the context of the current study. In this respect, COR pertains to the protection of individual resources instead of gaining opportunities due to the higher importance of existing resources, where gaining can be perceived as harmful. In contrast, individuals invest in certain resources for future protection (e.g., against loss and to recover a loss) [30,42–44]. This theory fits in the current context as it highlights how talented

employees can show turnover or lack of engagement when their personal resources are jeopardized by a crisis (i.e., the COVID-19 pandemic). The current study conceptualizes the notion of COR concerning the pandemic that caused vivid changes in work settings as noted earlier (e.g., downsizing, reduced wages, etc.), which can direct skilled employees towards leaving their jobs or seeking alternatives to ensure the conservation of personal resources. Disruptions in the hotel sector during the pandemic are regarded as an influential factor on employees in this industry, which is focused in this research from the perspective of managers in control of talent management strategies. COR describes personal protection of resources that is more important when facing health and work challenges (during the pandemic), which can prompt turnover, dissatisfaction, and low engagement that, combined with inadequate professional resources (provided by organizations) can lead to loss of talent [43,44]. Therefore, this theory highlights the importance of talent management strategies in providing a workplace where skilled staff can feel valued and cared for and where the organization aids them in maintaining their resources that were depleted during the pandemic. Both theories promote employee engagement, particularly those with high-demanding jobs (i.e., hospitality and hotel sectors) [45–47]. Based on the premises of the aforementioned theories, this research aims to yield results that can be beneficial for managers of hotels in the Middle East in the post-pandemic era as well as scholars interested in this subject. It is also important to note that the theories in this study (i.e., JD-R, human capital science and theory and COR) have been used in the data analysis process for deriving preliminary codes.

## 3. Methodology

Based on the aims and scope of the current research, an inductive qualitative approach was deemed appropriate as the endeavor is to provide in-depth data from managers in the hotel industry to understand better the current status of talent management practices as well as well-being and engagement aspects for employees. Particularly this study follows similar studies in the literature in terms of conducting interviews [48–50]. The study's themes were qualitatively developed from the data using inductive content analysis that followed a specific framework (1) specified the content for analysis; (2) definition of categories (themes) for analysis; (3) rules and criteria for coding (based on theoretical setting); (4) coding the data based on the criteria; and (5) conduct data analysis. Qualitative content analysis [51] fits the current research as it aims to obtain in-depth data that can be systematically treated. This research used in-depth interviews with human resource managers and CEOs in hotels to elicit themes and patterns. Following this procedure for transcripts, themes and patterns were identified and categorized based on similarity and dissimilarity. Accordingly, over a span of 3 months from December 2021 to March 2022, a total of thirty-seven (37) semi-structured interviews were conducted at the managerial level (departmental and HR). The setting of the interviews included various relevant themes that are further explained in the following passages. Job Demands, resources provided by HR and conservation of resources were used as the basis for establishing interviews in terms of context (themes that were discussed with participants).

### 3.1. Interview Procedure

Semi-structured interviews were conducted with 37 hotels across the Middle East (i.e., Lebanon, United Arab Emirates, Egypt, Jordan, Bahrain, Qatar, Saudi Arabia, Turkey and Oman. The inclusion criteria were that hotels were all five-star, located in the capital cities of the aforementioned countries, and were willing to participate. Accordingly, 22 hotels were from Lebanon and the United Arab Emirates (11 each), 4 from Egypt, 3 from Jordan, 2 from Bahrain, 2 from Qatar and 2 from Saudi Arabia, and Turkey and Oman (1 each). Interviews were conducted in English, as all managers were bilingual or multilingual. The hotels selected in this study are all from the Middle East and, thus, share a regional strategic plan applied in all regional branches. The targeted individuals in the current study hotels are the managers and, more specifically, HR managers. Nonetheless, the

sample also included general managers, CEOs and HR directors. Prior to data collection, relevant permissions were obtained from the directors and CEOs of the selected hotels. Interviews lasted 45–60 min on average and were held online through zoom meetings, WhatsApp and direct calls (based on participants' preference). Interviews were conducted, recorded and transcribed by the first author and supervised regarding conversation quality by the second author. Each interview contained 5 themes related to COVID-19's firsthand influence on hotels' human resources development and talent management strategies. The interviews were conducted individually to guarantee privacy, and all records were deleted after transcription to ensure anonymity. Necessary time was given to participants based on availability and willingness to participate. The theoretical setting of the research was taken into consideration, as well as the specified sample population, quality of dialogues and relevant analytical approaches [52–54]. The transcriptions were given to the interviewees to approve its content and provide any additional comments, upon which they were reread (by both researchers) numerous times to assure the accuracy of terms used by participants and the definitions/synonyms used in coded data to fit the academic narrative. The themes were constructed using transcriptions and are developed based on both expected and unexpected codes. The coding frame was applied to the transcripts in the study using QSR NVivo. Data analysis was conducted by the first researcher and supervised by the second author. To ensure the study's rigor and consistency, a freelance expert was consulted regarding the analysis procedure of interviews and coding, which increased the adequacy of codes and the aforementioned criteria regarding the inductive content analysis. An "intercoder agreement" was reached with a 0.91 coefficient, which confirms the reliability of procedures.

### 3.2. Respondents' Profile

To address the aims and scope of the current study, only managerial-level individuals were selected for interviews. Selected hotels were all five-star and in the private sector, due to confidentiality and ethical means, remain anonymous. The age range of participants was from 39 to 5,8, with 14 female and 23 male interviewees. Participants comprised 26 HR managers, 7 general managers, 2 HR directors (cluster), and 2 CEOs. These managers were directly involved in the processes of HR and talent management in their organizations, contributing to the strategic development of practices and approaches in the hotel regarding their staff. Only 3 participants held bachelor's degrees, while others held master's degrees in different areas (e.g., economics, HR, business, tourism, and marketing). Only one manager had 5 years of experience, with others having 7 to 18 years of relevant work experience and current positions. No personal or sensitive data were obtained nor remained in the final data. The original records were deleted, and initial data with identifiers were returned to the participants.

## 4. Results

The themes of questions were developed regarding the influence of the COVID-19 global pandemic on talent management, development of HR strategies, and employees' engagement among Middle Eastern hotels (e.g., Based on your experiences, how do you evaluate employees' attitudes towards their job during COVID-19 Pandemic? In your opinion, due to new world conditions and the COVID-19 Pandemic, what would be the new employee skills requirements of the hospitality industry? What kind of skills and behaviors do you see as talent? What is done in your company to retain your talent pool? What are the challenges of keeping your skilled staff? Can you tell us about the benefits of having talented staff?). Using inductive content analysis as the analytical technique [55–57] of qualitative data, concepts and themes can be derived to address the research objectives. In this respect, the aforementioned themes are shown in Table 1.

### 4.1. Theme 1—Evaluation of Employee's Attitude and Engagement

There were positive and negative aspects regarding the evaluation of employee's attitudes during the COVID-19 pandemic. Managers from the Marriott group noted that "*Employees were accepting new working conditions; they slowly adapted to the new situations, as challenging as they were, they tried to be as flexible as possible to perform efficiently*". Employees exhibited good working practices by being able to handle multiple tasks and working under pressure. Wyndham group managers reported, "*They showed loyalty to their work, understanding the pandemic situation and its effects by working hard, cooperating and communicating with customers*". Employees showed creativity and courage during their work, ensuring customer satisfaction. Employees were developing working conditions by following up strategies and effective protocols to support management and build mutual relationships. Managers from Accor group of hotels mentioned that "*Employees were respecting health and safety measures by applying all the necessary precautions such as hand sanitisers, mandatory masks, social distancing and following all the hygienic standards*".

One of the hotels in Lebanon stated that during the COVID-19 pandemic, they needed to be closer to their colleagues more than ever and ensure welfare strategies were in place so colleagues felt safe. "*A White Glove Service was introduced to protect guests and employees, and a back to work script was launched to guide HR teams in ensuring they had everything covered*", where business had been disrupted. Another Hotel also stated that in all cases, precautions were always taken regarding sanitizing, wearing face masks at all times and maintaining social distancing and applying new ventilations systems (e.g., MERV). "*When some of the staff tested positive with COVID-19 and the on-site staff doctor took care of them immediately and followed up with all remaining staff treating each case separately*" said managers from the Double Tree hotel group. This helped motivate them towards the organization and to encourage engagement with their jobs as care and support were provided.

According to interviewees, the engagement level of employees dropped vividly during the pandemic and especially in the first few months as there was no clear guideline from policies by the government or the organizations. It was reported that compared to the months before the pandemic (close to January 2019), there was a massive decrease in the engagement at work (including online meetings), when the pandemic occurred.

"*Due to the lockdown, employees became insecure about their jobs; they were afraid of being fired at any point, losing their jobs. They were anxious and living under the ongoing stress of the pandemic but, over time, adapted to the situation*". There was a decline and disruption in business revenue with a direct threat to the international economy due to a global health crisis. Employees faced struggles, when they felt uncertainty about their jobs, which lowered their engagement levels. Hilton hotel managers noted that "*Some employees were struggling to adapt to the new working conditions and were resistant to change*". Hotels across the Middle East tried to find optimum solutions by "*Leading with Care*" to overcome the negative impacts.

All hotels were closed to curb the spread of the virus, which posed a challenge for both the employer and employees. Four Season hotel managers reported that "*Firstly, the hospitality industry did not allow employees to work online and so the first step was ensuring our team members were kept safe in their homes and feeling secure by getting their salaries on time. Secondly, we worked on disinfecting the hotel and implementing the requirements for protecting employees and guests. Thirdly, online training was arranged, updating the team members on the international best practices for preventing COVID-19. Before any approvals were given to open the hotels, the team members were trained on how to work during this crisis while we provided the tools needed, the change in work conditions and the preventions. During the evaluation of COVID-19, we ensured employees and managers were following the instructions, criteria and prevention methods of the Ministry of Health that has become part of the evaluation system*". A hotel in Egypt states that employees were very positive before the pandemic. As hospitality was one of the most impacted industries by the pandemic, "*they didn't feel safe anymore*". According to interviewees, the engagement of employees was at high levels before the pandemic, especially in the months leading to the COVID-19 outbreak. However, as the

pandemic started, a vivid decrease was noted in employees' engagement. They feared they would lose their jobs and not receive income, but with all the measures taken with "*Lead with Care*", they felt they could continue their journey. Hence, they embraced it and were cooperative with their superiors. Providing information about the situation, the disease, and policies made employees more aware of how well the guests should be treated and how important their engagement with the job is. It was also noted that managers unanimously mentioned the positive impact of talented employees on their customers' experience due to high engagement levels, positive attitudes, and taking initiatives to enhance provided services "*we can see that in areas where our most skilled staff work, our customers tend to give be more satisfied . . . they* [customers] *also mention their names* [staff] *in evaluation surveys that we use*". Hence, it can be interpreted that the presence of talented employees can lead to higher customer satisfaction rates as the ultimate outcome.

### 4.2. Theme 2—Talent Management Strategies (Employees' Recruitment and Skills)

A Jordanian hotel stated that talented employees were performing great; they were trained and educated in all aspects of talent management. Employees were willing to learn and adapt to the COVID-19 pandemic, which played an effective role in developing employees skills. Now more than ever, there is a need for high adaptability to the changing world caused by the pandemic. Managers in Marriott hotels said, "*Resilience, grit, ability to multitask, safety and security play vital roles in our employee skills requirements*". While such traits are desired in the recruitment process, companies have developed practices to increase the skill level of their talented staff, who are in the company. Hotels in Lebanon and UAE noted that new employees adapted and showed a strong capability to handle stress, a high level of safety vision, and multifunctional to have a team with high productivity. Talented employees have effective skills such as developing tech, flexibility and adaptability, communication and emotional intelligence, creativity and innovation as well as customer service skills. "*There is a need for multi-skilled employees working in some areas in the hotels*" said a manager in the Four Season hotel group. In line with talent management strategies, managers noted that skillful employees are highly essential and are sought after by HR departments as the core strategy. This is alongside their endeavors to provide training and development programs that can improve staff abilities and "*develop talent in our employees*". This shows that not only hotel managers (and HR) sought skilled employees to be recruited but also emphasized on improving skills possessed by their talented staff in integrated strategic planning that entailed the recruitment process (e.g., required skills), and HR talent management strategies (e.g., developmental activities).

Other hotels in Lebanon and Turkey recall that employees should develop their skills and build adaptability and resilience swiftly for the pandemic and endure their social and emotional skills. "*Recruiting employees depends on their attitude, ability to learn and abide to the organizations' law, capable of following the safety measures, precautions, instructions, and to be both confident and brave*". Managers also noted that "*Good communication*" and "*confidence*" are the traits for an employee with long future in this sector. The ability to change to virtual business and openness to education are also the main traits for employees willing to have a new role in the organization. These skills were taken into consideration for recruiting new employees as well as providing incentives for existing talents, who also received training specific to their roles.

In addition to the positive feedback (from employees and customers) regarding deployed measures given by a selection of hotels, there was some negative feedback, which managers tried to find solutions for. The lockdown affected the customers' satisfaction and employees' performance, resulting in talented employees leaving their jobs and having to downsize the hotel (turnover and economic pressure). Some of the hotels were not providing sufficient training to employees. "*The focus should have been on virtual business recruitment, online interviews and hiring vaccinated employees*". Direct feedback from a hotel in Qatar is that "*hotels should focus on virtual recruitment and more digital training strategies. The new generation is tech-savvy and spends more time on technology than any other generation.*

*Training should be simplified and tailored for specific needs rather than handling long classroom trainings*". Employees' skills (whether new recruits or existing ones) directly link with the company's HR strategy for attracting and developing talented staff. In this sense, interviewees noted that for implementing strategies for retaining talent, skills were evaluated, and new practices (e.g., training) were introduced in the recruitment section. The degree to which a new recruit exhibited skills related to their job became a more dominant factor for their recruitment.

Reports from Bahrain and Qatar expressed that "*first, the feasibility of the candidate to travel should be checked*", considering the restrictions implied. "*Second, the intention of employment, as we are looking for longevity considering a wide range of competitors*". A Lebanese hotel added that "*the quality of people changed in terms of employees and customers especially, in the last 2 to 3 years*". "*Costumers have different motivations and expectations. Many talented employees are not here anymore. Entry-level employees are hard to find*". According to Lebanese hotels, "*COVID-19, the revolution and the Beirut port explosion*" impacted the industry negatively, and the lockdown affected the service and customer satisfaction. This also impacted their recruitment and training for staff as the society had to deal with political, economic and crisis challenges. This can be linked to COR theory, as individuals are more likely to conserve and protect their resources during a crisis than seek new opportunities. Additionally, the degree to which the employee is fit in the organization was evaluated in the HR department as a new strategic process for talent recruitment. Lastly, competitive analysis was more focused (on marketing and HR teams) to benchmark new ideas and strategies to maintain employees and focus on talent development.

### 4.3. Theme 3—Talent Evaluation and Retention

Talent evaluation is the overall process that includes aspects of hiring, training, retention, and rewards [39]. A brief overview is given before delving into further details. As the hospitality sector focuses on improving its talent management efforts in the progressing global economy, primarily four stages emerge that must be followed for proper assessment of talent [58]. The first stage is the assessment of new talent itself. As such, this stage is where talent management needs to be most effective as noted by Marriott managers "*when we find a new talent, we start by recruiting them for a specific role, but as they grow and show their talents, we often end up giving them a new position that matches their abilities . . . where they can be most helpful*". Expanded to all HR practices, when staff are recruited, managers seek to provide a better fit for their staff from quarterly or annual evaluations of performance, which can lead to promotions, bonuses, and offering new developmental paths that aim to retain the employee. In addition, HR carries a vital role in recognizing the job demands and available resources for employees to conduct their tasks. It was noted that Wyndham managers stated that "*we have weekly meetings with HR . . . if HR team does one thing wrong, it will have many bad effects that can cause financial loss for the company, and that is never what we want*". Hilton managers noted that "*our best people are in the HR department, where they can choose staff that meet the standards we are looking for . . . . if you choose wrong people in HR, the whole company will be filled with bad systems and staff*". The second stage is evaluation and assessment in a systematic manner. Accuracy alone does not make up for the assessment portion. Instead, a more accurate measure of competency is needed to give holistic and detailed data on the employee. The traditional recruitment agencies and unstructured interview approach only do so much to provide a solid insight. It was mentioned by Four Season manager[s] that "*for every position we have many applicants that it becomes difficult to choose from . . . sometimes most skilled people are not fit . . . .often compensation is the key and other times personal issues that hold them from joining us*". Managers noted that the evaluation process is constant and focuses more on employees who show commitment, engagement, and talent. While some departments (e.g., kitchen) evaluate their staff every month, others (e.g., marketing) have assessments quarterly. Managers reported that for directing resources (training, equipment, etc.) to talented employees, the ongoing assessment processes are definitive. Especially during the pandemic, where job demands shifted and increased,

HR managers focused on previous assessments to select the employees they sought to retain. Evaluations were made with certain employees regarding their needs for handling the new situation. This encompassed both physical and psychological aspects of the job (linked to the JD-R model and human capital and science theory). These findings relate to the aforementioned theories as they show the vitality of recognizing and tending to the needs of employees during a crisis as they become more vulnerable and, thus, require more support and care from their organizations.

The third stage is the provision of continued development. Performance appraisals make up a large portion of feedback and development; as noted by Accord managers, "*all managers are involved in decisions based on incentives, appraisals, and training . . . we make sure that each department gets what they need to motivate our best staff*". Nevertheless, the main problem here is that appraisals are usually summative and do not look beyond current performance levels to highlight how future performance may be affected. During the pandemic, new approaches were introduced that focused on the overall performance of the employee since their recruitment. This enabled managers to distinguish high-performing individuals who might not have been identified through old measures. This was because engagement decreased during the pandemic and many staff did not have the necessary resources (personal and professional) to perform well. The fourth and last stage constitutes the promotion of the right talent. Instead of promoting already identified individuals, newer employees need to be focused on being encouraged and helping the company grow [59,60]. Marriott group states that "*if we promote an employee, they will come out of promotion cycle for the year . . . we want everyone to have a chance for promotion and we know how important it is to keep employees in the same level of treatment*". Wyndham managers also stated that "*we have 5 different promotional and incentive programs for our employees . . . we are planning to increase the number so that we can include more employees in such programs and to make them happier and create a healthy competition for good performance*". Considering the changes in job demands during the pandemic, the aforementioned programs and incentives that focus on all employees and their improvement in the company have been reported to be an effective strategy for retaining talent. Managers reported that as they demanded that staff comply with new policies and the situation (e.g., lowered wages), they recognized the importance of providing alternative means so that employees (especially talented ones) do not show turnover. Managers also observed a rise in the need for enhanced communication and provision of information during the pandemic. Managers across all departments were tasked with delivering information to employees regarding policies, changes in processes, demands, and changes in the availability of resources.

Talent retention aims to encourage employees to continue working with the company for a more extended period. These employees are essential to the company's success as they lead it towards future success. The cost of being replaced and the ramifications of losing them can be detrimental to a business in the hospitality sector [60,61]. Profitable businesses must come up with and implement strategies that focus on talent retention by involving employees in the process of decision-making. They must also be involved in processes where they are helped with recognizing their competence and role in the company's success, implementing a reward system for good performance, and creating opportunities for growth and development [62,63]. Managers across different groups of hotels noted the importance of retention. Accord managers said that " *. . . imagine having a talent that works for some time, you invest in them, you provide training, you give them knowledge and develop their understanding of hotel management. Not only do you not want to lose them, but if you do, they may be hired by your competitors, who may then intentionally or unintentionally share confidential information about us or use their expertise to improve their own business . . . we do our best to keep employees . . . the longer they stay with us, the more we want to keep them*". This implies that evaluation processes are vital for decisions made by managers concerning their talented staff. Furthermore, when uncertainty rises (i.e., during the pandemic), managers in hotels need to implement appropriate approaches to ensure that demands and resources are recognized, shortcomings are identified, and retaining talent is emphasized.

### 4.4. Theme 4—Difficulties of Retaining Talented Employees and the Role of Managers

The COVID-19 pandemic harmed recruitment management by finding difficulties in paying salaries, increasing payroll, retaining talented employees and running a reward system and assessment. The lockdown forced the hotels to freeze recruitment, minimize the number of employees, and decrease the number of working hours, which led to giving incomplete salaries by using the cost-cutting approach strategy [64]. The difficulties of retaining talented employees had a negative impact on economical situations, specifically in the hospitality industry. Due to this, it was suspected that there would be a recession and financial restrictions by losing high calibre talent employees. As a result, talented employees will vanish and immigrate to find a better career progression abroad [65]. It was reported in UAE that "*when the pandemic happened, all managers were invited to have discussions on how to manage our employees, especially those with talent and skill or high potential . . . . we had meetings every day until we reached some decisions...we gathered information from each manager and tried to contact each employee separately to give them a solid answer from their managers about their work and future*".

According to our participants, "*usually*", talented employees can be retained by giving them chances to grow and create a good environment through bonuses and motivational approaches. Talented employees' loyalty during the pandemic was based on financial treatment. Moreover, developing employees' skills via training and raising awareness sessions can increase the sense of belongingness by stabilizing vivid opportunities and retention programs, increasing loyalty. A problem arose that some hotels have "*limited resources and investment in rewards, training development and recognition programs, difficulties in affording to promote and training expenses, insufficient compensation plans, limited options for growth, and work-life imbalance*" (e.g., Turkey, and Lebanon). A manager in Turkey said that "*we gave options to our mid-managers to do what they see best for their best staff . . . incentives, bonuses, and many good ideas came from this practice, and we were able to satisfy a big number of our employees*".

According to managers in the Marriott group of hotels in UAE, Oman, and Saudi Arabia: "*We didn't have difficulties retaining talented employees because we restructured our hotel, and with talented people, we expanded the roles. We have over 5 hotels under our wing, so instead of having 1 training manager in each branch, I promoted my training manager to a cluster training manager. Hence, we enriched their job description*". This resulted in career growth for all employees by giving them more responsibilities and roles, which led employees to believe this was a chance for growth rather than just pressuring them with more tasks. "*This was our main approach to retain the talent*".

It was also expressed by the Hilton group of hotels that the skilled staff will not stay further in a company with an unclear vision for the future. The managers feared the loss of talent in getting people working in Lebanon, Oman, and Saudi Arabia. Thus, the choices/options of attracting the right people, working in the right position at the right place and retaining them vanished.

> "*I do not consider developing the skills and motivating the staff to be in the Talent management section or HR. This is more related to the company as one entity. It is survival that matters and if you want talent to stay, you should keep them motivated and wanting the job or, better, the company. They should want to be future managers.*"

Respondents from Turkey, UAE, Egypt, Bahrain and Jordan noted in their opinion that difficulty in retaining people was low as the majority were either laid off or being paid a half salary, with the general strategy being "*keep your job and survive*". In the opinion of managers in Saudi, the impact is that "*after the business picks up and everything returns to normal, those who feel they were not treated fairly will leave the hotel to other competitors for sure*". Employees' loyalty during the pandemic either "*went up or down based on how they were treated financially and otherwise*".

In Lebanon, it was stated that "*too many difficulties*" faced organizations during the COVID-19 Pandemic. The entire hospitality sector was confronting a high recession, and the

organizations were losing money. They expressed that for HR Managers, it becomes more difficult to raise salaries and increase the payroll, retain talented employees, run a reward system, and conduct a thorough assessment. To survive this situation, HR are mixing jobs, giving employees fewer working hours, introducing "*work from home*" opportunities, minimizing employee numbers, seeing more turnover during the lockdown, vivid stress and fear and different schedules and performance. Regarding retaining talent during the pandemic, it was said that "*yes, it affects the business in a harmful way, but we are looking forward to crossing safely from this stage and being better prepared for future crises*". Another top-level manager in Jordan stated that "*owners of the hotel allowed us to hire consultants and specialists like psychologists to help with decisions for our staff . . . we used many resources to make sure that we are doing the right thing for our best employees . . . .I oversaw the processes and hired counselling services for those who had problems during the pandemic*". The aforementioned findings show managers' essential role regarding talent management strategies used during the pandemic.

*4.5. Theme 5—Advantages of Talented Employees and Organizational Culture*

The advantage of selected talented employees is that they can easily adapt to new working conditions by coping with and handling stress [8,14,66]. Talented employees are productive and have many skills, including technical and soft skills, proactivity, and multitasking. Talented employees can effectively fulfil requirements and overcome obstacles and remote work arrangements [67,68].

Reportedly it was noticed that talented employees have a good financial impact on the industry by reaching "*sales target*" and "*guest satisfaction*". They sustain a good reputation and personal plan development. During the pandemic, talented employees "*respected and complied with health and safety measures and carried out weekly PCR tests*". Managers in Jordan noted it, Turkey, Oman and UAE that changes in an organizational setting (culture) enabled their talented employees to show more engagement and positive behaviors. In this respect, it was reported that "*we gave them [talented staff] an opportunity to be heard and imply their ideas. We asked them what they see as negative in the workplace that we can change . . . we created surveys and held meetings with best employees and were surprised by the sheer amount of input they had*". Marriott managers reported that "*our company's culture surrounds ethics and good performance . . . we have open communication with all staff. We always have our top staff involved in meetings that can influence other employees*". Four Season managers said that "*we like innovation in our company and invite our staff to come up with new ideas to make the hotel a better place for them and our customers . . . we spend most of our time at work, so we always appreciate innovative ideas from our staff to make it feel like home*". Managers in Accord group of hotels reported that "*we always encourage team-building and delegation . . . our best employees are the ones who help others grow and can handle teamwork . . . during the pandemic, we had a big community of employees that were each other's mental support*". These findings show the importance of organizational culture that enables talented staff to be engaged with the work and promotes positive communication, innovation, and teamwork.

Talented employees have compensation plans to feel secure, valued and motivated about their job in an appraisal performance. Since the sample criteria of this study purposively aimed at high-quality hotels, it was observed across the data that employees whose talent was noted by their managers were offered higher salaries and better coverage options (e.g., medical insurance, incentives, extra holidays, etc.). a manager from Marriott—UAE stated the following with regard to financial importance of talent:

> "*We retained our talented employees and gave them more responsibilities and bigger roles which had a good financial impact. We saved the cost of hiring more people and our top employees were multitasking. They became new faces of the company in their manners. We have success stories; we were promoting ours when other hotels laid off people*".

## 5. Theoretical Understanding

The core aim of this research was to contribute to the extant literature on talent management and, by extension, HRM and business (i.e., hotel) management. To achieve this, in-depth data was obtained through interviews reported above. Current results show that managers in the hotel industry across the Middle East know, recognize and value the talent in their respective organizations. This shows that due to the extent of competitiveness and rivalry in this sector, five-star hotels have established an organizational environment where talent is appreciated and specific tactics and strategies are used to promote engagement, satisfaction, and retention. The findings show that managers carry an important role in fostering a workplace for talent. In addition, HR departments can improve engagement and ensure retention by implementing practices that focus on the needs of employees during uncertain times of a crisis. Evaluation and assessment processes are key determinants in recruiting talent and providing practices that pave the way for their development in the company and, thus, encourage the talented individual to remain in the organization. The current findings show the importance and applicability of the JD-R model, human capital and science theory, and COR as theoretical contributions to the literature that suggests managers in hotels need to recognize the difficulties of jobs assigned to staff, shifting demands during the crisis, and the tendency of individuals to preserve their resources to face the situation better. As the literature shows gaps in talent management during the pandemic and strategies that managers deployed to handle the crisis, these findings can benefit practitioners and scholars.

In this respect, the current research expands the theoretical application of the JD-R model as managers enable their employees to exhibit and express their ideas and skills. It was noted that in-person (one-on-one or group) meetings were held with the department and HR manager(s) and talented employees. Each employee would be heard regarding the conduct of their tasks, providing resilience and autonomy to talent. Similarly, the necessary information was provided to staff alongside delivering policies and regulations, which are considered signals that managers provide to their talented staff. Managers specifically addressed the shortcomings of resources (e.g., decreased revenue which led to reduced wages). Additionally, recognition of new demands (e.g., longer hours, remote working, and restricted access to different sections) were highlighted for employees within the context of the JD-R model. While some reduced the salaries and had limited options to provide (e.g., training, coverage, remote work, etc.), stronger groups in terms of the number of hotels and scale offered alternative initiatives for their talented employees to retain and further develop into more crucial roles. This was established through various training programs (knowledge or activity-based). Managers were encouraged by the organization to enhance efficiency in their own initiatives, enabling the implementation of various strategies. As managers endeavored to provide adequate information to their staff, JD-R model and human capital and science theory can be linked to current findings as it shows the importance of meeting the needs of employees for retaining talented staff in the hotel sector.

In the premise of human capital science and theory [22] the needs of talented employees after their recognition were addressed by their managers and the HR team. This especially was noted among marketing, sales, technical, and logistics departments at and throughout all levels (domestic or regional). In the context of this theory, the human capital requires an active HR department with an emphasis on sustainable development and resilient organization that provides care for their employees during a crisis. This similarly addresses the application of the JD-R model [36,37], as JD-R encompasses physical and psychological aspects of the job, the two theories show that for better managing talent during a crisis, it is imperative that their needs are tended by the organization while shortcomings of resources are addressed. Changes in job demands are adequately delivered to staff. This is because with high demands and particularly low resources (decreased revenue which led to reduced wages, downsizing, and new demands for different working hours and compliance with restrictions) during the pandemic, human capitals' retention is highly dependent

on organizational strategies that recognize employees' needs and aim to enhance their work settings. This enhancement can be achieved when both the demands and resources of jobs are clearly defined by managers, and staff are adequately informed about limits and uncertain aspects of their jobs during a crisis. Five-star hotels across the Middle East show positive feedback in service quality, performance and customer experience, which are imperative for organizational success [39,40]. Current results show that combined premises of the JD-R model, human capital science and theory, can be implied within the context of enhancing work environment through appropriate practices that focus on engagement, satisfaction, and retention of talented staff in hotels [69,70]. As these theories focus on meeting the needs of employees, their application is salient in the current context.

In the premises of COR theory, individual resources (e.g., against loss and to recover a loss) [30,43,71] are involved, which is linked to the JD-R model implies that employees in hotels need to conserve their resources during crises. This implies that to maintain performance while ensuring their futures are secured, HR departments in hotels should deploy appropriate strategies that entail physical and psychological domains of an employee. This becomes more important for retaining talented staff as their resources should suffice in terms of meeting immediate needs (i.e., during a crisis) and long-term needs (e.g., to cope with stress, workload, future career development, and adequate compensation plans). Hotels implemented new evaluation methods, increased communication with their staff, elaborated on changes in policies, and held routine meetings to provide support and counselling to their employees. These were deployed to reduce stress caused by uncertainty. Organizations can recognize the COR theory in this context as individuals can withhold their skills to face the crisis better. In this sense, practices and activities that support staff become essential. Hotel managers faced challenges in recruiting new talent due to the pandemic, which can be linked to conserving resources rather than seeking opportunities. Additionally, lowered engagement among staff can be linked to COR as the decline in engagement was observed by managers during the pandemic. Hotel managers used training, incentives, developmental plans, and new assigned role as different tactics to aid their staff with reducing stress, providing care and support, and focusing on their wellbeing (e.g., mentoring, counseling, incentives, and healthcare support) during the pandemic. As hotels are considered high-demanding jobs [46,47], their talented employees require to be ensured of their careers and motivated. As noted in the current results, hotels as companies have increased the tasks of certain employees, which in return have enabled them to show newly found skills, leading to promotions, more important roles, and inclusion in decision-making processes. Cognitive and physical resources are used in the hotel sector for completing tasks. Hence, when resources are limited and demands are high, employees are more prone to conserve their mental and physiological resources. Importantly, during the pandemic, various risks were posed to individual lives, which further encouraged the conservation as mentioned above. Therefore, the current results show that managers should pay close attention to the implications of the theories used in this research. Accordingly, this research pertains to the theoretical setting deployed for its conduct.

### 5.1. Practical Implications

Due to the nature of data gathered in this research, a number of practical implications can be interpreted that can be beneficial for managers across the Middle East in the service, hospitality, tourism and hotel sectors. For instance, online courses and flexible schedules can help develop specific skills. This states that the service sector can implement educational courses via HR departments to maintain engagement and increase the interactions between the company and employees, considering several work departments directly interact with customers in hotels. As found in the results of this research, having talented employees can increase customer satisfaction as an ultimate outcome. This is interpreted through the understanding of interactions among employees and customers when the staff exhibit high engagement levels, are supported by their organizations and managers and are equipped with necessary tools (physical and psychological aspects of the job) that

enable them to be heard and take initiative. The role of leaders and HR departments in establishing an environment where talented staff are cared for and both personal and professional domains of their lives are fostered in their organizations is essential. Through adequate strategic planning, hotel managers can not only prepare for future crises. Still, they can ensure that their customers are provided with a high-quality service that is vital for organizational success. It is imperative that hotels provide alternative compensation plans to reduce anxiety related to job security and future plans. Longer contracts can be signed to exhibit a sense of commitment and further motivate talent and reduce turnover. HRM departments play a major role in this context and its interconnectedness with leadership and other departments. Furthermore, organizational support [9,10,12,72] as a vital element for employee engagement should be considered, where hotels can implement the initiatives above and countless other programs specific to the area, budget and size to ensure that their talented employees will remain and not only help in the development of the company but also develop professionally and personally.

*5.2. Limitations and Recommendations*

Despite the obtained result and their benefits, several limitations restrain the conduct of this study: (a) there are a limited number of studies that address this subject and thus, comparing different contexts is not feasible to empower the support for such arguments. Hence, future studies, regardless of location, can develop the current understanding of the subject of talent management and particularly in the post-pandemic era; (b) this study focused on managers and their input regarding the talent management context. Future studies can address employees as they are essential in the service industry and provide data regarding their preferences and needs in the aftermath of the pandemic; (c) as qualitative data was gathered, generalizability of results can be diminished regardless of the fact that data was obtained from several countries, which prompts future quantitative studies to further analyze current findings; (d) although process of data collection was time-consuming, longitudinal studies can provide a thorough understanding of how different initiatives impact retention, engagement, and motivation of talented employees in service sector; (e) the relationship among different departments of organizations and the linkage between employees and their managers was not regarded in this research, which can be addressed by scholars interested in the subject; and (f) cultural studies can be highly beneficial to examine differences and similarities among diverse workforces of hotel and hospitality industries.

**Author Contributions:** Conceptualization, F.H. and H.Ö.; methodology H.Ö.; Data Collection, F.H.; writing, F.H.; review and finalization, F.H. and H.Ö. All authors have read and agreed to the published version of the manuscript.

**Funding:** This research received no external funding.

**Institutional Review Board Statement:** Not applicable.

**Informed Consent Statement:** Participants were contacted several times before the Interviews and verbal consent was given.

**Data Availability Statement:** The coded map of the data in this study are available on request from the corresponding author.

**Conflicts of Interest:** The authors declare no conflict of interest.

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
