# Peer review of "Sustaining Human Resources through Talent Management Strategies and Employee Engagement in the Middle East Hotel Industry"

_sustainability, doi:10.3390/su142215365_

Round 1

Reviewer 1 Report

Thank you for the opportunity to review the manuscript titled “Sustaining Human Resources through Talent Management  Strategies and Employee Engagement in the Middle East Hotel  Industry”.

I believe that while the topic is interesting and the work done is commendable a further significant work is required to clearly demonstrate the need for the study, its relationship with other research in the area of employee engagement and wellbeing and finally the contribution of this project.

The manuscript also needs a thorough proofreading.

Introduction

In my opinion the Introduction could be significantly improved in order to clearly state the motivation for the research, research objectives and research questions. Currently it reads as very repetitive and it is hard to delineate the key points. It would be good to see in the introduction the overall statement of the problem – perhaps related to the disruptions induced by Covid-19 and implications for the workforce and their engagement and wellbeing. Then it would be good to narrow down this problem to a particular aspect (I suppose how it influenced employees in the hospitality industry), briefly present what we know about this problem and where the gaps, which the research is going to address, are. Would be good to state the research questions here as well in addition to the research aim.

It might be useful to have a look at Grant and Pollock (2011) for some tips about writing the Introduction.

Line 30 I would not call Talent management a technique.

Line 40 “High quality staff and resources” – not clear which resources

Line 29- 45 are essentially saying the same thing. It might make sense to delete repetition and compress the text.

Lines 54 – 58 – not sure what is meant by this sentence – might need re-writing.

Lines 59-60  - Would be good to add references to the definition of Engagement

I failed to see signalling theory as relevant to the discussion of talent management here– this might need some unpacking.

Literature Review

The introduction states the following aim of the study: “Accordingly, this study aims to explore and gain a deep understanding of talent management strategies among hotels across the Middle East and how they encourage employee  engagement in the tourism industry during the Covid-19 pandemic, and its implications in the aftermath of the pandemic”. I would suggest to craft the literature review around this aim to highlight what we know about talent management practices and engagement in relation to disruptions and/or resilience. Also, the literature review needs to highlight the gaps in our knowledge.

Although the choice of theoretical framework seems relevant it is largely left to the reader to connect these theories to the motivation for the study and study aims. The theoretical frameworks need proper presentation and their relevance to research aims need to be clearly demonstrated. Why do employees need to conserve resources (e.g. some studies on stress in hospitality industry), why JD-R framework is relevant? What perspective does it give to the study?

That employees’ engagement leads to positive organisational outcomes is well established  in the literature, there is a need to justify why engagement in Covid-19 and in post-Covid times is essential. Also, the link between the HRM practices and engagement is well addressed in the literature  – there is a need to articulate the gap the study is going to address. It would be good to bring theoretical perspectives into this discussion.

Lines 177-178: Based on the premises of aforementioned theories, this research aims  to contribute to the current understanding of their embeddedness within the context of  hotels in the Middle East in the post-pandemic era – Is this also the aim of the study? I did not see how it was addressed further in the Discussion (Theoretical Understandings). Whose embeddedness is not clear.

Lines 117- 120 – not sure what the sentence means.

The section 2.1. mentions that different theories are used. I was wondering if they can be introduced in this section and the choice of theoretical perspectives justified upfront. It might also be useful to explain in which capacity they were used in data analysis (e.g. preliminary codes were derived from them).

Lines 122-136 – probably belong to introduction more than to the Literature Review

Methodology

Why were participants selected from several countries? What are the consequences of this decision? How are these countries comparable?

In which language were the interviews conducted? How and by whom the translation was done?

Lines 213-214:  Theoretical setting of the research was taken into consideration as well as specified sample  population, quality of dialogs and relevant analytical approaches – What does this sentence mean?

Line 216 – Accuracy of what?

Line 217:  It is not clear how framework can be constructed based on transcriptions – Did the authors mean themes identified through coding?

Line 218: Does it mean that first some interviews were coded, then coding scheme developed and applied to the rest? How many interviews were coded to develop a coding scheme?

Who participated in the data analysis?

Providing transcribed data to participants is commendable – have they made any changes? How these changes were treated?

 It might be suggested to present the demographic data in a table. Also, the authors might want to discuss why participants were selected to represent different categories of hotel management (e.g. Hr managers, General managers, CEOs). To which extent are their accounts comparable?

Analysis and conclusion

I would suggest to restructure this section and present the data in the way that the themes speak to the research objectives and questions. I believe the main emphasis should be on HR strategies and their outcomes for engagement. It would also be good to see how the theories informed the data analysis – this might be reflected in themes (e.g. demands, resources developed by HR, resource conservation).

Lines 233-234: To ensure the study's rigour and consistency, linguists and subject experts analysed  and evaluated 37 interviews criteria were met. Not clear what this means. Who did the data analysis?

Lines 283 – 294 – Is it a quote from the interview? If so, it needs to be properly attributed.

Lines 294 - A hotel in Egypt states that employees were very positive and 294 because hospitality was one of the most impacted industries by the pandemic, they didn’t feel safe anymore - the sentence does not make sense. After that there is again a first-person speech – can it be properly attributed? These occurrences of the first-person accounts in the text need to be checked and corrected.

Theme 2 – Only paragraph three perhaps reflects the theme.

Theme 3 – Not clear what comes from the literature, what comes from the analysis and what represents authors’ speculations.

Lines 408-409 – What does this sentence mean?

Theoretical understanding

This section needs to be enhanced and the findings put back in the context of the literature and theory in a more structured and clear way. It was hard to understand how the findings contribute to theories mentioned and what they add to the literature on talent management and employee engagement. I also believe that the key conclusions with respect to the study aims need to be more articulated.

Lines 456-457 – Not clear what is meant here. In which sense is the theory application extended?

Lines – 470-472 – what is meant is not clear.

Grant, A. M., & Pollock, T. G. (2011). Publishing in AMJ—Part 3: Setting the Hook. Academy of Management Journal, 54(5), 873-879. doi:10.5465/amj.2011.4000

Author Response

Open Review 1:

(x) I would not like to sign my review report

( ) I would like to sign my review report

English language and style

(x) Extensive editing of English language and style required

( ) Moderate English changes required

( ) English language and style are fine/minor spell check required

( ) I don't feel qualified to judge about the English language and style

Yes         Can be improved             Must be improved          Not applicable

Is the content succinctly described and contextualized with respect to previous and present theoretical background and empirical research (if applicable) on the topic?

( )            ( )            (x)          ( )

Are all the cited references relevant to the research?

( )            (x)          ( )            ( )

Are the research design, questions, hypotheses and methods clearly stated?

( )            ( )            (x)          ( )

Are the arguments and discussion of findings coherent, balanced and compelling?

( )            ( )            (x)          ( )

For empirical research, are the results clearly presented?

( )            ( )            (x)          ( )

Is the article adequately referenced?

(x)          ( )            ( )            ( )

Are the conclusions thoroughly supported by the results presented in the article or referenced in secondary literature?

( )            ( )            (x)          ( )

Comments and Suggestions for Authors

Thank you for the opportunity to review the manuscript titled “Sustaining Human Resources through Talent Management  Strategies and Employee Engagement in the Middle East Hotel  Industry”.

I believe that while the topic is interesting and the work done is commendable a further significant work is required to clearly demonstrate the need for the study, its relationship with other research in the area of employee engagement and wellbeing and finally the contribution of this project.

The manuscript also needs a thorough proofreading.

Response: thank you for your thorough review and constructive comments. We have proofread the manuscript in its entirety. Applied changes are highlighted in yellow in the revised version of the manuscript.

Introduction

In my opinion the Introduction could be significantly improved in order to clearly state the motivation for the research, research objectives and research questions. Currently it reads as very repetitive and it is hard to delineate the key points. It would be good to see in the introduction the overall statement of the problem – perhaps related to the disruptions induced by Covid-19 and implications for the workforce and their engagement and wellbeing. Then it would be good to narrow down this problem to a particular aspect (I suppose how it influenced employees in the hospitality industry), briefly present what we know about this problem and where the gaps, which the research is going to address, are. Would be good to state the research questions here as well in addition to the research aim.

It might be useful to have a look at Grant and Pollock (2011) for some tips about writing the Introduction.

Line 30 I would not call Talent management a technique.

Line 40 “High quality staff and resources” – not clear which resources

Line 29- 45 are essentially saying the same thing. It might make sense to delete repetition and compress the text.

Lines 54 – 58 – not sure what is meant by this sentence – might need re-writing.

Lines 59-60  - Would be good to add references to the definition of Engagement

I failed to see signalling theory as relevant to the discussion of talent management here– this might need some unpacking.

 Response: we highly appreciate your details review and have adjusted the introduction section accordingly.

Literature Review

The introduction states the following aim of the study: “Accordingly, this study aims to explore and gain a deep understanding of talent management strategies among hotels across the Middle East and how they encourage employee  engagement in the tourism industry during the Covid-19 pandemic, and its implications in the aftermath of the pandemic”. I would suggest to craft the literature review around this aim to highlight what we know about talent management practices and engagement in relation to disruptions and/or resilience. Also, the literature review needs to highlight the gaps in our knowledge.

Response: thank you for your valuable comments. We have adjusted literature review section.

Although the choice of theoretical framework seems relevant it is largely left to the reader to connect these theories to the motivation for the study and study aims. The theoretical frameworks need proper presentation and their relevance to research aims need to be clearly demonstrated. Why do employees need to conserve resources (e.g. some studies on stress in hospitality industry), why JD-R framework is relevant? What perspective does it give to the study?

Response: we have revised the theoretical section in accord with your comments.

That employees’ engagement leads to positive organisational outcomes is well established  in the literature, there is a need to justify why engagement in Covid-19 and in post-Covid times is essential. Also, the link between the HRM practices and engagement is well addressed in the literature  – there is a need to articulate the gap the study is going to address. It would be good to bring theoretical perspectives into this discussion.

Response: we have added more information in this regard.

Lines 177-178: Based on the premises of aforementioned theories, this research aims  to contribute to the current understanding of their embeddedness within the context of  hotels in the Middle East in the post-pandemic era – Is this also the aim of the study? I did not see how it was addressed further in the Discussion (Theoretical Understandings). Whose embeddedness is not clear.

Response: we have revised this section.

Lines 117- 120 – not sure what the sentence means.

The section 2.1. mentions that different theories are used. I was wondering if they can be introduced in this section and the choice of theoretical perspectives justified upfront. It might also be useful to explain in which capacity they were used in data analysis (e.g. preliminary codes were derived from them).

Lines 122-136 – probably belong to introduction more than to the Literature Review

Response: we have provided similar data in the introduction section. Additionally, we have revise section 2.1 and added the usage of theories in data coding process. 

Methodology

Why were participants selected from several countries? What are the consequences of this decision? How are these countries comparable?

In which language were the interviews conducted? How and by whom the translation was done?

Response: we have provided the information in this regard to the methodology section.

Lines 213-214:  Theoretical setting of the research was taken into consideration as well as specified sample  population, quality of dialogs and relevant analytical approaches – What does this sentence mean?

Response: this is the criteria that is used within the thematic network approach to ensure appropriateness of procedures.

Line 216 – Accuracy of what?

Response: we have added the information in this regard.

Line 217:  It is not clear how framework can be constructed based on transcriptions – Did the authors mean themes identified through coding?

Response: sorry for the writing error and thank you for pointing it out. As you mentioned ‘framework’ is supposed to be themes. We have changed the term.

Line 218: Does it mean that first some interviews were coded, then coding scheme developed and applied to the rest? How many interviews were coded to develop a coding scheme?

Response: the term ‘remaining’ was mistakenly written there. We have removed it as all coding processes were done for all interviews.

Who participated in the data analysis?

Response: We have added this information to the section.

Providing transcribed data to participants is commendable – have they made any changes? How these changes were treated?

Response: no additional comments was added. Participants were asked to check terminologies and reports, which were confirmed. Also, based on coding, any synonym or similar words were asked from participants to ensure that transcripts have not changed any meanings.

 It might be suggested to present the demographic data in a table. Also, the authors might want to discuss why participants were selected to represent different categories of hotel management (e.g. Hr managers, General managers, CEOs). To which extent are their accounts comparable?

Response: We have provided the information regarding the direct influence of these managers on HR and talent management strategies in their organizations. Regarding the table for demographics, as there are a very limited amount of data obtained in this regard, the table would be very small and empty. Hence, we decided to report the profile of participants. We hope that this satisfies your concern.

 Analysis and conclusion

I would suggest to restructure this section and present the data in the way that the themes speak to the research objectives and questions. I believe the main emphasis should be on HR strategies and their outcomes for engagement. It would also be good to see how the theories informed the data analysis – this might be reflected in themes (e.g. demands, resources developed by HR, resource conservation).

Response: we have provided more information regarding the theories and themes. However, the main focus of the research is talent management practices that address engagement. Hence, the themes and interviews surround this matter and not all HR strategies (e.g. internal programs that are not made for talent specifically). Hence, we have focused on the elements and information that specifically address talent management.

Lines 233-234: To ensure the study's rigour and consistency, linguists and subject experts analysed  and evaluated 37 interviews criteria were met. Not clear what this means. Who did the data analysis?

Response: we have revised the sentence to clarify.

Lines 283 – 294 – Is it a quote from the interview? If so, it needs to be properly attributed.

Lines 294 - A hotel in Egypt states that employees were very positive and 294 because hospitality was one of the most impacted industries by the pandemic, they didn’t feel safe anymore - the sentence does not make sense. After that there is again a first-person speech – can it be properly attributed? These occurrences of the first-person accounts in the text need to be checked and corrected.

Theme 2 – Only paragraph three perhaps reflects the theme.

Theme 3 – Not clear what comes from the literature, what comes from the analysis and what represents authors’ speculations.

Lines 408-409 – What does this sentence mean?

Response: in the original submitted file we have made the responses clear but for some reason it does not show. We have put the responses in quotation marks.

Theoretical understanding

This section needs to be enhanced and the findings put back in the context of the literature and theory in a more structured and clear way. It was hard to understand how the findings contribute to theories mentioned and what they add to the literature on talent management and employee engagement. I also believe that the key conclusions with respect to the study aims need to be more articulated.

Lines 456-457 – Not clear what is meant here. In which sense is the theory application extended?

Lines – 470-472 – what is meant is not clear.

 Response: we appreciate your critical notes that have led to increased quality of our manuscript. We have made adjustments throughout the file in accord with your recommendations.

Grant, A. M., & Pollock, T. G. (2011). Publishing in AMJ—Part 3: Setting the Hook. Academy of Management Journal, 54(5), 873-879. doi:10.5465/amj.2011.4000

Reviewer 2 Report

In my opinion, it is necessary to add illustrative material to the presented article.

It is good to present the results of the research done graphically. That way, there will be greater clarity.

Author Response

Open Review 2:

( ) I would not like to sign my review report

(x) I would like to sign my review report

English language and style

( ) Extensive editing of English language and style required

( ) Moderate English changes required

(x) English language and style are fine/minor spell check required

( ) I don't feel qualified to judge about the English language and style

Yes         Can be improved             Must be improved          Not applicable

Is the content succinctly described and contextualized with respect to previous and present theoretical background and empirical research (if applicable) on the topic?

(x)          ( )            ( )            ( )

Are all the cited references relevant to the research?

(x)          ( )            ( )            ( )

Are the research design, questions, hypotheses and methods clearly stated?

( )            (x)          ( )            ( )

Are the arguments and discussion of findings coherent, balanced and compelling?

( )            (x)          ( )            ( )

For empirical research, are the results clearly presented?

( )            (x)          ( )            ( )

Is the article adequately referenced?

(x)          ( )            ( )            ( )

Are the conclusions thoroughly supported by the results presented in the article or referenced in secondary literature?

(x)          ( )            ( )            ( )

Comments and Suggestions for Authors

In my opinion, it is necessary to add illustrative material to the presented article.

It is good to present the results of the research done graphically. That way, there will be greater clarity.

Response: we highly appreciate your comments and suggestion to increase the quality of our research. As the coded data was analyzed in a specific approach, visualization of data (network illustrations) are not feasible. However, we have clarified the responses from managers and the interpretations of their reports in a clear manner after adjustments that will aid the reader in following the narrative of the research. The changes are highlighted in yellow throughout the text. We hope that this satisfies your concern.

Reviewer 3 Report

This article explores talent management strategies to encourage employee engagement in the Middle East tourism industry during and after Covid-19 and further explores the impact of employee engagement on customer satisfaction. The study was conducted qualitatively, where open-ended questions were asked to 37 managers through semi-structured interviews. The study involved HR managers from numerous Middle East hotels in countries such as Lebanon, the United Arab Emirates, Egypt, Jordan, Bahrain, Qatar, Saudi Arabia, Turkey, and the Sultanate of Oman. Most hotels are rated five stars, while others are rated four stars. Research topics were qualitatively developed from the data using inductive content analysis deployed in QSR NVivo. The results showed that by implementing appropriate talent management strategies, hotel staff engagement and job satisfaction could be increased. The Covid-19 pandemic has shown that it is necessary to set realistic goals to support and retain talented employees effectively. Lack of resources and investment in talent management strategies (such as reward systems) can lead to losing talented employees. The results are helpful for both scientists and leaders of the hotel industry in the Middle East region.

Despite the satisfactory quality of the article, some shortcomings need to be corrected.

  1. It should be grounded on why 37 interviews are enough to have valuable results.
  2. Providing more details about respondents will increase the quality of research.
  3. It is recommended to visualize the results of interviews.
  4. It is recommended to rename section 4, Analysis and Conclusions, to Results.
  5. It is recommended to include the Discussion section to compare obtained results with other research.
  6. The scientific novelty of the study should be highlighted.

The main recommendation of the review is to include a visualization of the results. It will increase the quality of the paper slightly.

Author Response

Open Review 3:

(x) I would not like to sign my review report

( ) I would like to sign my review report

English language and style

( ) Extensive editing of English language and style required

(x) Moderate English changes required

( ) English language and style are fine/minor spell check required

( ) I don't feel qualified to judge about the English language and style

Yes         Can be improved             Must be improved          Not applicable

Is the content succinctly described and contextualized with respect to previous and present theoretical background and empirical research (if applicable) on the topic?

( )            (x)          ( )            ( )

Are all the cited references relevant to the research?

(x)          ( )            ( )            ( )

Are the research design, questions, hypotheses and methods clearly stated?

( )            ( )            (x)          ( )

Are the arguments and discussion of findings coherent, balanced and compelling?

( )            (x)          ( )            ( )

For empirical research, are the results clearly presented?

( )            ( )            (x)          ( )

Is the article adequately referenced?

( )            (x)          ( )            ( )

Are the conclusions thoroughly supported by the results presented in the article or referenced in secondary literature?

( )            (x)          ( )            ( )

Comments and Suggestions for Authors

This article explores talent management strategies to encourage employee engagement in the Middle East tourism industry during and after Covid-19 and further explores the impact of employee engagement on customer satisfaction. The study was conducted qualitatively, where open-ended questions were asked to 37 managers through semi-structured interviews. The study involved HR managers from numerous Middle East hotels in countries such as Lebanon, the United Arab Emirates, Egypt, Jordan, Bahrain, Qatar, Saudi Arabia, Turkey, and the Sultanate of Oman. Most hotels are rated five stars, while others are rated four stars. Research topics were qualitatively developed from the data using inductive content analysis deployed in QSR NVivo. The results showed that by implementing appropriate talent management strategies, hotel staff engagement and job satisfaction could be increased. The Covid-19 pandemic has shown that it is necessary to set realistic goals to support and retain talented employees effectively. Lack of resources and investment in talent management strategies (such as reward systems) can lead to losing talented employees. The results are helpful for both scientists and leaders of the hotel industry in the Middle East region.

Despite the satisfactory quality of the article, some shortcomings need to be corrected.

It should be grounded on why 37 interviews are enough to have valuable results.

Response: We appreciate your comments and time to review our research, which can only increase its quality. We have had access to 40 managers based on our network and specific criteria of the research. In addition, this was the point of data saturation, where repetitive comments were noted.

Providing more details about respondents will increase the quality of research.

Response: we have provided the extracted demographic characteristics as well as the criteria that was used for data collection. Due to confidentiality issues, we did not gather personal info beyond position, gender, and age. We hope this satisfies your concern.

It is recommended to visualize the results of interviews.

Response: we have provided a clear narrative of responses in the section.

It is recommended to rename section 4, Analysis and Conclusions, to Results.

Response: we have renamed the section in accord with your constructive comments.

It is recommended to include the Discussion section to compare obtained results with other research.

The scientific novelty of the study should be highlighted.

The main recommendation of the review is to include a visualization of the results. It will increase the quality of the paper slightly.

Response: we appreciate your comments that have increased the overall quality of our paper. The manuscript has been edited in its entirety and changes are highlighted in yellow.

Dear Authors,
I have carefully reviewed the reviewers’ comments and revisions of the manuscript title “Sustaining Human Resources through Talent Management Strategies and Employee Engagement in the Middle East Hotel Industry”. The current manuscript is well written. In its current form, this paper is on the quality standards of an academic paper. So, I suggest minor revisions. The minor revisions are expressed below.

English: Writing needs to be improved must. Writing doesn't mean English or grammar mistakes only. The way of expression is also important. Indeed, using appropriate words, logical sequence, and constructing a good pitch is all important. It is also suggesting the authors use research-oriented language.

Response: we highly appreciate your constructive comments and valuable time that you put to review our paper. We have made changes throughout the text.

Abstract: The abstract of this study needs more clarity. Explain more the research methods and analysis techniques that were used in this study. It also suggests you conclude and explain the implication of the study.

Response: we have edited the abstract in accord with your suggestion.

Introduction: It is also suggested to provide a more rational background of the gap of the study and research questions. As well as explain your research questions in detail, what research will you do? I also suggest to the authors, in the last paragraph of the introduction, explain the structure of the paper. It is suggested that the authors carefully read and follow the below-mentioned studies' research methods parts.
l http://doi.org/10.2147/PRBM.S204662
l https://doi.org/10.1016/j.energy.2021.122765
Research Methodology: The research methodology of this study also needs further explanation. The detail is mentioned below.
i. Which research approach did the authors use in this study? (Develop a sub-heading with the title of the Research approach and explain what research approach you used in this study and why you use this research approach specifically in your study.)
ii. Below mentioned studies will help you to improve your methodology.
iii. https://doi.org/10.1007/s11356-021-16441-6
iv. Asghar, Muhammad Zaheer, et al. "Adoption of social media-based knowledge-sharing behaviour and authentic leadership development: evidence from the educational sector of Pakistan during COVID-19." Journal of Knowledge Management ahead-of-print (2022).
Conclusion: This conclusion section also needs further explanation. I suggest you integrate it with the literature and results of the study.

Response: dear editor, we highly appreciate your comments and have used the recommended citations to enhance the quality of our paper. All changes are highlighted in yellow throughout the text. We hope that this meets your expectations.

Round 2

Reviewer 1 Report

Thank you for the opportunity to review the revised version of the manuscript titled “Sustaining Human Resources through Talent Management  Strategies and Employee Engagement in the Middle East Hotel  Industry”

Abstract

One of the objectives is to show how engagement impacts of customer satisfaction – Could you please briefly present the findings pertaining to this objective in the Abstract as well? Also, I haven’t seen these findings in the Results section, so probably it makes sense to present them there as well.

Lines  21-22: The Covid-19 pandemic showed that realistic targets needs to be set to effectively maintain employees and by retaining talented employees – this sentence needs re-writing.

Introduction

I believe that the Introduction will benefit from re-writing. First, it is very repetitive. It is not clear what the key motivation for the study and what contributions the authors are going to make. Why do we need to know about the talent management within and after Covid-19? Why is it important to look at talent management during disruptions? (please note that it is not very clear form the whole paper whether it addresses TM in or after Covid-19). What is the perspective from which the authors are looking at talent management during disruptions – to which theory are the authors going to contribute?  Why is this theoretical perspective relevant and useful? Finally, what are the contributions this paper makes?

Lines 30-32 – the sentence needs re-writing, as it is not clear what the authors mean here.

Lines 35-37 – the sentence needs rewriting.

Lines 37 – 39 – needs reference for increased investment

Lines 52- 54 – needs reference for issues

Literature review

If the authors clearly identify the problem this study addresses, then the Literature Review could be written around this problem. This way it will become much more specific.

Section 2.1. The key gaps in the TM literature this study is going to address are not clear. The only gap which is salient is the lack of studies in Middle East.  Why are they important? Due to the cultural context? How is this context then appreciated in this study? Due to something else? What?

The importance of looking at TM after Covid-19 is stated (after or within?) but again needs a better justification with the help of the literature. What do we know about the talent management during and after disruptions? How do disruptions impact talent management? The authors need to clarify the perspective from which they look at the problem. Why is this perspective useful?

Section 2.2 – reads very repetitive with the same points raised many times. I believe it needs tightening up. The authors need to explain clearer with the support from the literature why the focus is on the employee wellbeing (it is hard to see where the wellbeing comes up in the Results section) in TM, as other factors can contribute to employee engagement. Wellbeing apperes here quite suddenly – if this is the key focus it needs to be presented earlier. In Results wellbeing did not come across very well, so the question is Why was it stressed in the Literature Review? There is also a need to unpack the JD-R framework and COR much more in relation to this study objectives and justify their use better. What does the use of these theories add to this study? What new will we get to know looking from that perspective about TM in disruptions?

Methodology

Lines 195-196:  Qualitative content analysis [54] fits the current researches textual data can be systematically treated – Not clear what it means.

Lines – 226 – 228 – Not clear what is meant by the accuracy of terms and definitions.

Lines 228-229 – there is a contradiction between inductive analysis and a coding frame application. There is a need to say how the coding frame was developed and applied in an inductive manner.

Results

Line 247-248 – There is a need to explain what this consultation included and why it was needed. Also, this statement is a part of Methodology section rather than results.

Lines 258-260 – How is the use of preliminary coding related to inductive analysis? The use of preliminary coding assumes deductive approach. The reference 58 describes the inductive content analysis process stating that it starts with the open coding (open coding does not come from the preliminary codes). Again, this should be moved to the Methodology section.

Table 1 – It is not clear how these themes speak to the research questions and how they are related to the theories. I would suggest to further consult the literature which discusses qualitative data analysis.

There is a need to code participants, so that the quotes could be correctly attributed without compromising participants’ confidentiality.

There is a need to present data in a meaningful way. I was wondering how the authors interpreted what they have heard during the interviews, otherwise we are reading a number of quotations and are left to make sense of them on our own. Quotations need to be accurately selected to support and illustrate the authors’ key points, but not to be the key content of the sections.

Lines 316-326 – do not reflect the Theme, they are not about talent management strategies. Same for lines 331—333.

Line 335 – Positive feedback to what?

The Theme 2 presentation ends by saying that changes in the environment affected HR strategies. The authors need to present the new strategies very clearly, cluster them in a meaningful way and use only a few quotes to support their conclusions. From the current presentation we can understand only few strategies (e.g. relying more on technology).

Theme 3 – it seems to be a part of TM strategies (retention strategies). Why is it a separate theme is not clear. I was wondering if the themes were named inaccurately.

Theme 3 – the whole presentation is very confusing. There are too many references to the literature and no quotes  so it is not clear what the findings from this study are. I have made same comment in the previous revision. If the authors see my comment irrelevant, could they please rebuttal in the response?

After Theme 3 the logic of the presentation is changed. I would be expecting to see the presentation of Theme 4 and 5 (The role of managers in managing talented employee and Organizational culture and talent management strategy). I was wondering if this change of the presentation logic could be explained and justified.

Line 404 – The Marriott group is mentioned – in the methodology the authors stated that the hotel names won’t be mentioned.

Lines 421 – 422 – What does this sentence mean? (I asked the same question in the previous review).

Difficulties in retaining employees is perhaps an interesting section which presents some retention strategies. Can these strategies be grouped?

I did not see any point in the theme (sub-theme?) ‘Advantages of talented employees’ as it does not provide any new insights about the advantages, the retention strategies mentioned in this section could be put in the relevant sections.

Overall, the Results section is very raw. I would probably suggest to reanalyse data to make sure that the analysis helps to respond to the research questions. Also, since the study uses some theories these theories need to come across in the analysis – usually in case of inductive analysis this happens at the stage of aggregation.

Theoretical understanding

First sentence – these contributions need to be unpacked very clearly in the paper.

Lines – 468-472 – Where did we see this in Results? There is a need to make sure that in the Discussion presented results are discussed.

Signalling theory did not come across well in the Discussion. The authors either need to strengthen it or to omit.

Current results imply that combining JD-R model, human capital science and theory and signaling theory can vividly impact on enhancing work environment through appropriate practices that focus on engagement and wellbeing of employees in hotels” – It is not clear what is meant by this sentence and how theories impact the work environment.

COR theory in the Discussion – we actually haven’t seen any examples of COR in the Results (see my previous comments about the need to link analysis to theoretical background). Hence, it is not clear where this recommendation comes from.

The use of HD-R framework seems relevant, but the data need to be reanalysed with respect to this framework to provide meaningful insights.

The research questions are related to the TM, employee engagement and customer satisfaction – the answers to these questions need to be clearly presented in the Discussion based on the findings presented in the previous section.

Thank you for the opportunity to review the revised version of the manuscript titled “Sustaining Human Resources through Talent Management  Strategies and Employee Engagement in the Middle East Hotel  Industry”

Abstract

One of the objectives is to show how engagement impacts of customer satisfaction – Could you please briefly present the findings pertaining to this objective in the Abstract as well? Also, I haven’t seen these findings in the Results section, so probably it makes sense to present them there as well.

Lines  21-22: The Covid-19 pandemic showed that realistic targets needs to be set to effectively maintain employees and by retaining talented employees – this sentence needs re-writing.

Introduction

I believe that the Introduction will benefit from re-writing. First, it is very repetitive. It is not clear what the key motivation for the study and what contributions the authors are going to make. Why do we need to know about the talent management within and after Covid-19? Why is it important to look at talent management during disruptions? (please note that it is not very clear form the whole paper whether it addresses TM in or after Covid-19). What is the perspective from which the authors are looking at talent management during disruptions – to which theory are the authors going to contribute?  Why is this theoretical perspective relevant and useful? Finally, what are the contributions this paper makes?

Lines 30-32 – the sentence needs re-writing, as it is not clear what the authors mean here.

Lines 35-37 – the sentence needs rewriting.

Lines 37 – 39 – needs reference for increased investment

Lines 52- 54 – needs reference for issues

Literature review

If the authors clearly identify the problem this study addresses, then the Literature Review could be written around this problem. This way it will become much more specific.

Section 2.1. The key gaps in the TM literature this study is going to address are not clear. The only gap which is salient is the lack of studies in Middle East.  Why are they important? Due to the cultural context? How is this context then appreciated in this study? Due to something else? What?

The importance of looking at TM after Covid-19 is stated (after or within?) but again needs a better justification with the help of the literature. What do we know about the talent management during and after disruptions? How do disruptions impact talent management? The authors need to clarify the perspective from which they look at the problem. Why is this perspective useful?

Section 2.2 – reads very repetitive with the same points raised many times. I believe it needs tightening up. The authors need to explain clearer with the support from the literature why the focus is on the employee wellbeing (it is hard to see where the wellbeing comes up in the Results section) in TM, as other factors can contribute to employee engagement. Wellbeing appears here quite suddenly – if this is the key focus it needs to be presented earlier. In Results and Discussion wellbeing did not come across very well, so the question is: Why was it stressed in the Literature Review? There is also a need to unpack the JD-R framework and COR much more in relation to this study objectives and justify their use better. What does the use of these theories add to this study? What new will we get to know looking from that perspective about TM in disruptions?

Methodology

Lines 195-196:  Qualitative content analysis [54] fits the current researches textual data can be systematically treated – Not clear what it means.

Lines – 226 – 228 – Not clear what is meant by the accuracy of terms and definitions.

Lines 228-229 – there is a contradiction between inductive analysis and a coding frame application. There is a need to say how the coding frame was developed and applied in an inductive manner.

Results

Line 247-248 – There is a need to explain what this consultation included and why it was needed. Also, this statement is a part of Methodology section rather than results.

Lines 258-260 – How is the use of preliminary coding related to inductive analysis? The use of preliminary coding assumes deductive approach. The reference 58 describes the inductive content analysis process stating that it starts with the open coding (open coding does not come from the preliminary codes). Again, this should be moved to the Methodology section.

Table 1 – It is not clear how these themes speak to the research questions and how they are related to the theories. I would suggest to further consult the literature which discusses qualitative data analysis.

There is a need to code participants, so that the quotes could be correctly attributed without compromising participants’ confidentiality.

There is a need to present data in a meaningful way. I was wondering how the authors interpreted what they have heard during the interviews, otherwise we are reading a number of quotations and are left to make sense of them on our own. Quotations need to be accurately selected to support and illustrate the authors’ key points, but not to be the key content of the sections.

Lines 316-326 – do not reflect the Theme, they are not about talent management strategies. Same for lines 331—333.

Line 335 – Positive feedback to what?

The Theme 2 presentation ends by saying that changes in the environment affected HR strategies. The authors need to present the new strategies very clearly, cluster them in a meaningful way and use only a few quotes to support their conclusions. From the current presentation we can understand only few strategies (e.g. relying more on technology).

Theme 3 – it seems to be a part of TM strategies (retention strategies). Why is it a separate theme is not clear. I was wondering if the themes were named inaccurately.

Theme 3 – the whole presentation is very confusing. There are too many references to the literature and no quotes  so it is not clear what the findings from this study are. I have made same comment in the previous revision. If the authors see my comment irrelevant, could they please rebuttal in the response?

After Theme 3 the logic of the presentation is changed. I would be expecting to see the presentation of Theme 4 and 5 (The role of managers in managing talented employee and Organizational culture and talent management strategy). I was wondering if this change of the presentation logic could be explained and justified.

Line 404 – The Marriott group is mentioned – in the methodology the authors stated that the hotel names won’t be mentioned.

Lines 421 – 422 – What does this sentence mean? (I asked the same question in the previous review).

Difficulties in retaining employees is perhaps an interesting section which presents some retention strategies. Can these strategies be grouped?

I did not see any point in the theme (sub-theme?) ‘Advantages of talented employees’ as it does not provide any new insights about the advantages, the retention strategies mentioned in this section could be put in the relevant sections.

Overall, the Results section is very raw. I would probably suggest to reanalyse data to make sure that the analysis helps to respond to the research questions. Also, since the study uses some theories these theories need to come across in the analysis – usually in case of inductive analysis this happens at the stage of aggregation.

Theoretical understanding

First sentence – these contributions need to be unpacked very clearly in the paper.

Lines – 468-472 – Where did we see this in Results? There is a need to make sure that in the Discussion presented results are discussed.

Signalling theory did not come across well in the Discussion. The authors either need to strengthen it or to omit.

Current results imply that combining JD-R model, human capital science and theory and signaling theory can vividly impact on enhancing work environment through appropriate practices that focus on engagement and wellbeing of employees in hotels” – It is not clear what is meant by this sentence and how theories impact the work environment.

COR theory in the Discussion – we actually haven’t seen any examples of COR in the Results (see my previous comments about the need to link analysis to theoretical background). Hence, it is not clear where this recommendation comes from.

The use of JD-R framework seems relevant, but the data need to be re-analysed with respect to this framework to provide meaningful insights.

The research questions are related to the TM, employee engagement and customer satisfaction – the answers to these questions need to be clearly presented in the Discussion based on the findings presented in the previous section.

Author Response

we highly appreciate your time and concern and constructive feedback that have led to significant improvement of our paper.

we have revised the manuscript in its entirety and hope that we have met your expectations and satisfied your concerns.

please find attached response file and revised manuscript file ( new changes are highlighted in cyan)

Reviewer 3 Report

Thanks for the authors for considering comments and recommendations. Now, the paper can be accepted

Author Response

Open Review 3:

English language and style

( ) Extensive editing of English language and style required
( ) Moderate English changes required
(x) English language and style are fine/minor spell check required
( ) I don't feel qualified to judge about the English language and style

Yes

Can be improved

Must be improved

Not applicable

Is the content succinctly described and contextualized with respect to previous and present theoretical background and empirical research (if applicable) on the topic?

( )

(x)

( )

( )

Are all the cited references relevant to the research?

( )

(x)

( )

( )

Are the research design, questions, hypotheses and methods clearly stated?

( )

(x)

( )

( )

Are the arguments and discussion of findings coherent, balanced and compelling?

( )

(x)

( )

( )

For empirical research, are the results clearly presented?

( )

(x)

( )

( )

Is the article adequately referenced?

( )

(x)

( )

( )

Are the conclusions thoroughly supported by the results presented in the article or referenced in secondary literature?

( )

(x)

( )

( )

Comments and Suggestions for Authors

Thanks for the authors for considering comments and recommendations. Now, the paper can be accepted

Submission Date

29 August 2022

Date of this review

05 Oct 2022 11:19:33

RESPONSE: we highly appreciate your time and concern and constructive comments that enabled us to improve the quality of our paper.

Round 3

Reviewer 1 Report

Thank you for the opportunity to review the revised version of the manuscript titled “Sustaining Human Resources through Talent Management  Strategies and Employee Engagement in the Middle East Hotel  Industry”

There are some good improvements made in this version.

However, the manuscript still needs a very significant editing. The key contributions of the study do not come across very well. There are many sentences which meaning is not clear. The theories still need to be used in a more nuanced manner. There is a need to try to avoid using theories in a prescriptive manner.

It seems to me that the main contribution of this paper is in identification of the talent management strategies, which were used during the pandemic to ensure employee engagement and productivity. This is of importance as, as the authors indicate, this is the time of higher job demands, low level of resources available for use (e.g. finance) to support employees. This needs to be clearly outlined in the Introduction, come across in Findings and discussed in the Discussion section in relation to the literature and theory.

The introduction may need to discuss the talent management in ‘normal’ times and show what we know about talent management strategies in crisis. If not much is known, this should be stated and indicated as a gap this study aims to be addressed.

“This research addresses a number of aspects in this regard that are 1) importance of employee engagement in hotel sector; 2) talent management as
pects (i.e. retention, evaluation, and overall hotel talent management strategies); 3) role of managers in terms of handling talent; and 4) setting of the organization in terms of talent management”  - hard to understand what it means. Perhaps it makes sense to delineate some of the key contributions this study makes. Need to discuss these contributions in the Discussion section.

“The Middle East has continued to show resilience in hotel performance indicators prior to occurrence of the global pandemic, which had severe impacts on all industries and hospitality and hotel sector particularly, leading to detrimental effects on talent management” - it is not clear what is meant by resilience in hotel performance indicators. Also, this sentence might need to be split into several sentences.

“This research argues that for better handling work outcomes of hotel employees in the post-pandemic era, talent management strategies and practices should be implemented with the aim of encouraging engagement and increasing job satisfaction, which can lead to positive organizational outcomes such
as, service quality, performance, customer satisfaction and increased revenue” – Why is this important  only in post-pandemic era? It is also not clear whether the study focuses on what happened within pandemic or after.

Literature Review

Section 2.1. – there is a strong focus on what this study aims to accomplish, however there is a need to review relevant literature here with regard to Talent management in ‘normal’ times and in crisis (Talent management strategies and engagement, etc.).

There are many gaps indicated. How do we know that these are the gaps? Could the authors please add relevant references?

Section 2.2. – This section reads very repetitive. I think that both sections 2.1. and 2.2. can be combined, which will allow to streamline the literature review. It would be good to use more literature to discuss the link between talent management, engagement, demands and resources.

“This study focuses on the extent of which employees are engaged with
their work, which is particularly important for the service industry due to constant interactions among staff and customers
” – This does not seem to be the focus of this study.

There is a need to show with the references to the literature how engagement is related to JR-D framework.

“This requires an active HR department with emphasis on sustainable development and enhancing work settings for employees.” – this sentence needs re-writing. Also, there is a need to use the literature to demonstrate the link between sustainable development and engagement.

Again, there is a need to clarify whether the focus is on talent management during crisis or after crisis. For example, in the Results section we read that the questions were about what was happening during pandemic, while in the Introduction and Literature Review the authors state that the study contributes to the understanding of TM strategies after pandemic. There is a need for some consistency here.

Methodology

“Job Demands, resources provided by HR, and conservation of resources were used as rules of creating codes in coding process and categorizing data (themes)” – it is not clear how “job demands, resources and conservation of resources could be used as rules.

“definitions/synonyms used in transcribed data to fit the academic narrative” – do the authors mean ‘codes’ as transcribing does not presume replacing any terms to fit academic narrative?

“The themes were constructed using transcriptions and contain both expected and unexpected codes” – themes do not contain codes, they are developed based on the initial codes.

“To ensure the study's rigour and consistency, a freelance expert was consulted regarding analysis procedure of interviews and coding, which increased the adequacy of codes, and aforementioned criteria regarding inductive content 301
analysis” – what is meant by consulting here? Did the impartial expert do own coding for further comparison – interrater reliability? What were the outcomes? How did the authors deal with disagreement in coding?

Results

“development of HR and employees’ engagement” – what is meant by HR engagement?

“Codes are based on the theoretical framework of the research that pertains to these element” – this statement contradicts the previous statement that the coding was inductive.

“From our interviews, differences in evaluation of employee’s attitude and customer satisfaction were found to coincide with the different nature of employee engagement.” – It is not clear what this sentence means

Theme 1 – I would suggest again to synthesise the findings here rather than provide multiple quotes. It would be good to reduce quotes to just 1-2 to illustrate the key interpretations made by the authors. What are the key findings about what happened to employees’ engagement during the pandemic? Was it changing? Was it stable but different from pre-pandemic state?

Theme 2 – How were skills related to recruitment strategies? Why skills and recruitment are aggregated in a single theme is not clear. Skills part is more related to expectations and training to develop skills. While recruitment strategies does not seem to focus on skills needs.

“In-house training, integration of departments in terms of approach
and strategic management, introduction of new technologies, and applying strict health measures were noted as key strategic changes among hotel
– not clear how this is related to the Theme 2.

Theme 3 – It is good that the authors made clear what their findings are demonstrating. However, it is not clear why recruitment is included in this theme as well. It would be quite important to ensure that final themes are distinct.

I would again suggest to the authors to review the themes and link them to the theoretical framework. So that we see job demands, resources and strategies as clear themes.

Theoretical understanding

It would be really good if the authors could first summarise their key findings. Then it would be useful to put these findings in the context of theories especially JD-R. The authors need to clearly illustrate the new demands, the resources which seized to exist or were reduced due to pandemic and new resources that were created. Then the strategies which were applied to create these new resources need to be outlined.  Perhaps it would be good to put all these relationships in the model.

I still don’t see signalling theory coming across well. It is used rather superficially and perhaps the manuscript would benefit if it is not mentioned. Or, if the authors want to use it, it needs to be used in a more nuanced manner. It would be good to identify which practices were sending which signals. Why these signals were needed? Signalling theory is related to the situation when the information is incomplete, perhaps this needs to be discussed and illustrated here.

“As managers endeavored to provide adequate information to their staff, signaling theory can be linked to current findings as it shows the importance of this matter for retaining talented employees in the hotel sector.” – It is unclear what is meant by ‘this matter’.

“In the context of this theory, the human capital requires an active HR department with emphasis on sustainable development and resilient organization that provides care for their employees during a crisis, which similarly addresses the application of JD-R model [39,40]. It is not clear what this sentence means. What does the application of JD-R model address?

“and aim to enhance their work settings.” – not clear what this means

“Current results show that combined premises of JD-R model, human capital science and theory, and signaling theory can vividly be implied within the context of enhancing work environment through appropriate practices that focus on engagement, satisfaction, and retention of talented staff in hotels [75,76]. – It is not clear what this sentence means.

COR – clearly show which strategies were developed to enable and support resource conservation. So far it seems that the COR theory is used in a prescriptive manner – what organisations need to do based on this theory. I do not see it to be well connected to the findings. Again, the authors may consider how to use this theory in a more nuanced manner. What are the new strategies that are used by organisations to diminish stress? Did we get to know for example that during pandemics organisations focused more on strategies that help to support wellbeing?

It would be good to clearly outline the key contributions of this study in the Theoretical understanding. How does it contribute to the theory and the literature?

Author Response

Open Review

( ) I would not like to sign my review report

(x) I would like to sign my review report

English language and style

(x) Extensive editing of English language and style required

( ) Moderate English changes required

( ) English language and style are fine/minor spell check required

( ) I don't feel qualified to judge about the English language and style

Yes         Can be improved             Must be improved          Not applicable

Is the content succinctly described and contextualized with respect to previous and present theoretical background and empirical research (if applicable) on the topic?

( )            ( )            (x)          ( )

Are all the cited references relevant to the research?

( )            (x)          ( )            ( )

Are the research design, questions, hypotheses and methods clearly stated?

( )            ( )            (x)          ( )

Are the arguments and discussion of findings coherent, balanced and compelling?

( )            (x)          ( )            ( )

For empirical research, are the results clearly presented?

( )            (x)          ( )            ( )

Is the article adequately referenced?

( )            (x)          ( )            ( )

Are the conclusions thoroughly supported by the results presented in the article or referenced in secondary literature?

( )            ( )            (x)          ( )

Comments and Suggestions for Authors

Thank you for the opportunity to review the revised version of the manuscript titled “Sustaining Human Resources through Talent Management  Strategies and Employee Engagement in the Middle East Hotel  Industry”

There are some good improvements made in this version.

However, the manuscript still needs a very significant editing. The key contributions of the study do not come across very well. There are many sentences which meaning is not clear. The theories still need to be used in a more nuanced manner. There is a need to try to avoid using theories in a prescriptive manner.

RESPONSE: Dear reviewer, we appreciate your comments and feedback. We have revised the manuscript in its entirety in terms of the meaning of sentences and more arguments on theories and embedded them in contribution to the literature in the research. The revised file with tracking changes enabled so you can see the edits. Extensive English language editing and proofreading were also done.

It seems to me that the main contribution of this paper is in identification of the talent management strategies, which were used during the pandemic to ensure employee engagement and productivity. This is of importance as, as the authors indicate, this is the time of higher job demands, low level of resources available for use (e.g. finance) to support employees. This needs to be clearly outlined in the Introduction, come across in Findings and discussed in the Discussion section in relation to the literature and theory.

RESPONSE: Dear reviewer, we have provided the recommended information in the introduction section as well as the discussion section. Tracking changes is enabled in this version for your consideration.

The introduction may need to discuss the talent management in ‘normal’ times and show what we know about talent management strategies in crisis. If not much is known, this should be stated and indicated as a gap this study aims to be addressed.

RESPONSE: Thank you for your feedback. Since the issue of TM in crises time is not fully examined. We have stated that the gap is being tackled in this research.

“This research addresses several aspects in this regard that are 1) the importance of employee engagement in the hotel sector; 2) talent management aspects (i.e. retention, evaluation, and overall hotel talent management strategies); 3) the role of managers in terms of handling talent; and 4) setting of the organization in terms of talent management”  - hard to understand what it means. Perhaps it makes sense to delineate some of the key contributions this study makes. Need to discuss these contributions in the Discussion section.

RESPONSE: We have added more explanation in this stage, and have provided more discussions accordingly in the final sections.

“During the global pandemic, which had severe effects on all industries and the hotel industry in particular, the Middle East continued to exhibit resilience in hotel performance.” - it is not clear what is meant by resilience in hotel performance indicators. Also, this sentence might need to be split into several sentences.

RESPONSE: Dear reviewer, we appreciate your detailed comments and have revised the passage accordingly.

“This research argues that for better handling work outcomes of hotel employees in the post-pandemic era, talent management strategies and practices should be implemented with the aim of encouraging engagement and increasing job satisfaction, which can lead to positive organizational outcomes such as, service quality, performance, customer satisfaction and increased revenue” – Why is this important  only in post-pandemic era? It is also not clear whether the study focuses on what happened within pandemic or after.

RESPONSE: We have revised this passage as well as the manuscript in its entirety to omit such statements and clarify meanings.

Literature Review

Section 2.1. – there is a strong focus on what this study aims to accomplish, however there is a need to review relevant literature here with regard to Talent management in ‘normal’ times and in crisis (Talent management strategies and engagement, etc.).

RESPONSE: We have revised the section.

There are many gaps indicated. How do we know that these are the gaps? Could the authors please add relevant references?

RESPONSE: We have added the references that gaps were noted from in the relevant paragraphs.

Section 2.2. – This section reads very repetitive. I think that both sections 2.1. and 2.2. can be combined, which will allow to streamline the literature review. It would be good to use more literature to discuss the link between talent management, engagement, demands and resources.

RESPONSE: We have merged the two sections.

“This study focuses on the extent of which employees are engaged with

their work, which is particularly important for the service industry due to constant interactions among staff and customers” – This does not seem to be the focus of this study.

RESPONSE: We have revised the sentence.

There is a need to show with the references to the literature how engagement is related to JR-D framework.

RESPONSE: We have further highlighted the references supporting this linkage.

“This requires an active HR department with emphasis on sustainable development and enhancing work settings for employees.” – this sentence needs re-writing. Also, there is a need to use the literature to demonstrate the link between sustainable development and engagement.

RESPONSE: We have omitted sustainable development as it falls beyond the scope of current study; and have revised the section accordingly

Again, there is a need to clarify whether the focus is on talent management during crisis or after crisis. For example, in the Results section we read that the questions were about what was happening during pandemic, while in the Introduction and Literature Review the authors state that the study contributes to the understanding of TM strategies after pandemic. There is a need for some consistency here.

RESPONSE: We have revised the manuscript and the meaning of statements. We hope that the understanding of TM during the pandemic can lead to a better understanding on strategies that can be used when facing future crises. Therefore, we have revised the introduction and discussion accordingly.

Methodology

“Job Demands, resources provided by HR, and conservation of resources were used as rules of creating codes in coding process and categorizing data (themes)” – it is not clear how “job demands, resources and conservation of resources could be used as rules.

RESPONSE: We have revised the sentence as it did not reflect the meaning and we thank you for pointing out this shortcoming “used as basis for establishing interviews in terms of context (themes that were discussed with participants)

“definitions/synonyms used in transcribed data to fit the academic narrative” – do the authors mean ‘codes’ as transcribing does not presume to replace any terms to fit the academic narrative?

RESPONSE: We have revised the sentence as you have correctly noted, we had to replace terms that were not a good fit for the academic narrative (e.g. slang, jargons).

“The themes were constructed using transcriptions and contain both expected and unexpected codes” – themes do not contain codes, they are developed based on the initial codes.

RESPONSE: We have revised the sentence and appreciate your constructive comments.

“To ensure the study's rigour and consistency, a freelance expert was consulted regarding analysis procedure of interviews and coding, which increased the adequacy of codes, and aforementioned criteria regarding inductive content 301

analysis” – what is meant by consulting here? Did the impartial expert do own coding for further comparison – interrater reliability? What were the outcomes? How did the authors deal with disagreement in coding?

RESPONSE: The consultant was involved in the process of coding to reach an intercoder agreement with the coefficient of 0.91, which implied that both coders had a satisfactory agreement in terms of coding procedures. We have added this info to the text.

Results

“development of HR and employees’ engagement” – what is meant by HR engagement?

RESONSE: We have revised the sentence as it missed punctuations, which changed the meaning.

“Codes are based on the theoretical framework of the research that pertains to these element” – this statement contradicts the previous statement that the coding was inductive.

RESPONSE: We have omitted the sentence thanks to your constructive comment.

“From our interviews, differences in evaluation of employee’s attitude and customer satisfaction were found to coincide with the different nature of employee engagement.” – It is not clear what this sentence means

RESPONSE: This sentence was left from the first draft. We appreciate your keen eye and have thus, removed the sentence.

Theme 1 – I would suggest again to synthesise the findings here rather than provide multiple quotes. It would be good to reduce quotes to just 1-2 to illustrate the key interpretations made by the authors. What are the key findings about what happened to employees’ engagement during the pandemic? Was it changing? Was it stable but different from pre-pandemic state?

RESPONSE: We have revised the section accordingly.

Theme 2 – How were skills related to recruitment strategies? Why skills and recruitment are aggregated in a single theme is not clear. Skills part is more related to expectations and training to develop skills. While recruitment strategies does not seem to focus on skills needs.

RESPONSE: We have revised the section accordingly. Skills and recruitment were combined as managers noted the link between the two factors. We have now edited the section to further elaborate on this matter.

“In-house training, integration of departments in terms of approach

and strategic management, introduction of new technologies, and applying strict health measures were noted as key strategic changes among hotel – not clear how this is related to the Theme 2.

RESPONSE: We have omitted this sentence.

Theme 3 – It is good that the authors made clear what their findings are demonstrating. However, it is not clear why recruitment is included in this theme as well. It would be quite important to ensure that final themes are distinct.

I would again suggest to the authors to review the themes and link them to the theoretical framework. So that we see job demands, resources and strategies as clear themes.

RESPONSE: We have revised the section accordingly.

Theoretical understanding

It would be really good if the authors could first summarise their key findings. Then it would be useful to put these findings in the context of theories especially JD-R. The authors need to clearly illustrate the new demands, the resources which seized to exist or were reduced due to pandemic and new resources that were created. Then the strategies which were applied to create these new resources need to be outlined.  Perhaps it would be good to put all these relationships in the model.

RESPONSE: We have revised the section, added a summary of findings and elaborated on resources and demands and strategies undertaken.

I still don’t see signalling theory coming across well. It is used rather superficially and perhaps the manuscript would benefit if it is not mentioned. Or, if the authors want to use it, it needs to be used in a more nuanced manner. It would be good to identify which practices were sending which signals. Why these signals were needed? Signalling theory is related to the situation when the information is incomplete, perhaps this needs to be discussed and illustrated here.

RESPONSE: In accordance with your suggestion, we have omitted this theory from the manuscript.

“As managers endeavored to provide adequate information to their staff, signalling theory can be linked to current findings as it shows the importance of this matter for retaining talented employees in the hotel sector.” – It is unclear what is meant by ‘this matter’.

RESPONSE: We have omitted signaling theory and revised the sentence.

“In the context of this theory, the human capital requires an active HR department with emphasis on sustainable development and resilient organization that provides care for their employees during a crisis, which similarly addresses the application of JD-R model [39,40]. It is not clear what this sentence means. What does the application of JD-R model address?

RESPONSE: We have revised the sentence.

“and aim to enhance their work settings.” – not clear what this means

RESPONSE: We have elaborated on this section.

“Current results show that combined premises of JD-R model, human capital science and theory, and signaling theory can vividly be implied within the context of enhancing work environment through appropriate practices that focus on engagement, satisfaction, and retention of talented staff in hotels [75,76]. – It is not clear what this sentence means.

RESPONSE: We have revised the sentence and omitted the signaling theory.

COR – clearly show which strategies were developed to enable and support resource conservation. So far it seems that the COR theory is used in a prescriptive manner – what organisations need to do based on this theory. I do not see it to be well connected to the findings. Again, the authors may consider how to use this theory in a more nuanced manner. What are the new strategies that are used by organisations to diminish stress? Did we get to know for example that during pandemics organisations focused more on strategies that help to support wellbeing?

RESPONSE: We have elaborated on this matter in the section in accordance with your suggestion.

It would be good to clearly outline the key contributions of this study in the Theoretical understanding. How does it contribute to the theory and the literature?

RESPONSE: We have elaborated in this regard according to your suggestion.

Submission Date

29 August 2022

Date of this review

28 Oct 2022 10:43:21
